# Zero-Shot Performance Prediction for Probabilistic Scaling Laws

**Viktoria Schram[1]**   **Markus Hiller[1]**   **Daniel Beck[2]**   **Trevor Cohn**[*][1]

[1]School of Computing and Information Systems, The University of Melbourne
[2]School of Computing Technologies, Royal Melbourne Institute of Technology
{v.schram, m.hiller, trevor.cohn}@unimelb.edu.au
daniel.beck@rmit.edu.au

## Abstract

The prediction of learning curves for Natural Language Processing (NLP) models enables informed decision-making to meet specific performance objectives, while reducing computational overhead and lowering the costs associated with dataset acquisition and curation. In this work, we formulate the prediction task as a multitask learning problem, where each task's data is modelled as being organized within a two-layer hierarchy. To model the shared information and dependencies across tasks and hierarchical levels, we employ latent variable multi-output Gaussian Processes, enabling to account for task correlations and supporting zero-shot prediction of learning curves (LCs). We demonstrate that this approach facilitates the development of probabilistic scaling laws at lower costs. Applying an active learning strategy, LCs can be queried to reduce predictive uncertainty and provide predictions close to ground truth scaling laws. We validate our framework on three small-scale NLP datasets with up to 30 LCs. These are obtained from nanoGPT models, from bilingual translation using mBART and Transformer models, and from multilingual translation using M2M100 models of varying sizes.

## 1 Introduction

Large language models are increasingly being employed across various research domains and industrial applications, requiring substantial computational resources not only for the training, fine-tuning, and testing of these models but also for their deployment, prompting, and generation processes. As model complexity and the number of processed tokens increase, energy demands and processing times grow extensively, resulting in high costs and significant environmental impact [1, 2, 3, 4, 5]. In this context, *performance prediction* has emerged as a promising research direction, aiming to alleviate these costs by providing informed guidance for model selection, data acquisition strategies, and computational requirements [6, 7, 8].

One important task in this field of research is learning and training curve prediction. The former evaluates the generalization ability of machine learning models as a function of a resource of interest, such as dataset size or model complexity, while the latter models the evolution of the loss over training iterations, epochs, or steps [9, 10, 11]. Early approaches to learning and training curve prediction relied on fitting parametric functional forms to observed performance metrics and extrapolating future outcomes [12, 13, 14, 15, 16]. Since then, particularly within the context of Bayesian optimization, various surrogate models, such as Gaussian processes [17], Bayesian neural networks [18, 19], and ensemble methods [20], have been employed for curve extrapolation tasks, providing uncertainty estimates alongside point predictions.

---

[*]Also at Google Australia

39th Conference on Neural Information Processing Systems (NeurIPS 2025).

Training curves also play a key role in formulating and validating scaling laws, which are empirical relationships that describe how model performance varies as a function of factors such as model size, dataset size, and compute budget [3, 4, 21, 22]. However, deriving these scaling laws is highly resource-intensive, as it requires training numerous models and conducting extensive experiments across different configurations to obtain a representative set of training curves.

The *core hypothesis* that we explore in this paper is that our Natural Language Processing (NLP) learning and training curve datasets exhibit hierarchical, specifically bi-level, structures. A key advantage of hierarchical models is their ability to incorporate prior knowledge through the explicit definition of the hierarchy's structure. This not only enhances interpretability but also facilitates the transfer of knowledge across different tasks and domains [23, 24, 25, 26, 27]. Hierarchical modelling approaches have found applications in areas such as hierarchical text classification, hierarchical clustering of words within a vocabulary, and hierarchical Bayesian methods that incorporate multiple layers of data or prior information, for example, in word segmentation tasks [28, 29, 30, 31], among others. However, hierarchical modelling in the context of learning and training curve prediction tasks has surprisingly not yet been explored.

We aim to bridge this gap by modelling learning curve (LC) prediction tasks in NLP as multitask problems, where each task comprises data organized in a two-layer hierarchical structure, as illustrated in Figure 1. An *additional hypothesis* is that these hierarchies are exchangeable. We demonstrate that by leveraging correlations among tasks, zero-shot prediction of training curves for scaling law research can be achieved at significantly reduced com-

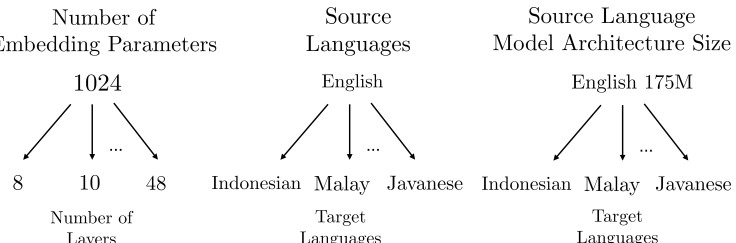

Figure 1: A task from each dataset illustrating the modelled two-layer hierarchy. (left) Learning curves assuming common prior information when the same number of embedding parameters is used. (middle) Learning curves for bilingual translation, assuming common prior information when translating from the same source language. (right) Learning curves for multilingual translation, assuming common prior information when the source language and model size are the same.

putational costs. Our framework employs latent variable multi-output Gaussian process models [24], which account for the hierarchical structure within each task as well as correlations across tasks. Our approach outperforms competitive baselines and leads to our *final hypothesis* that this model can be effectively used for scaling law prediction due to its strong zero-shot performance. This approach not only provides a principled mechanism for uncertainty quantification but also enables the derivation of probabilistic scaling laws via Monte Carlo simulation. Furthermore, by incorporating an active learning strategy, the uncertainty of the predictions can be substantially reduced. To demonstrate the generality of our data modelling approach, we extend this framework beyond scaling law prediction to performance prediction tasks in bilingual and multilingual translation across models of varying capacities. We show that, compared to baseline methods, leveraging hierarchical structure and task correlations yields significant improvements in zero-shot performance prediction, particularly for small-scale datasets. In the following, we will refer to training and learning curves as learning curves.

## 2   Related Work

**Learning Curves.**   Early studies on learning curve (LC) extrapolation analyzed functional forms, including power laws and exponential functions, among others [12, 16, 32, 33, 34, 35, 36]. LC trends have been extrapolated for larger training dataset sizes [9, 12, 32, 37, 38] or used to define early stopping criteria [39]. In the context of Bayesian optimization, surrogate models such as Gaussian processes with exponentially decaying kernels [17], Bayesian neural networks [18], or product kernels [19] have been employed for LC extrapolation. More recently, Transformer-based prior-data fitted networks have demonstrated competitive performance in LC extrapolation tasks [40]. Our work differs in that we explore and exploit existing hierarchical structures in small-scale NLP LC datasets.

**Zero-Shot Prediction.**   Zero-shot learning refers to a model's ability to make confident predictions for test data that were not encountered during training [41]. In NLP, zero-shot prediction has been applied to transferring models to new tasks [42, 43, 44], performing sentiment analysis in zero-shot

multilingual settings [45], and addressing cross-domain adaptation tasks [46], among others. In contrast, Bayesian optimization methods in hyperparameter optimization learn LC trends across various hyperparameter configurations and can predict outcomes for unseen configurations [18, 20]. Our work is most related to Klein et al. [18], where a Bayesian neural network with a basis-function layer performs zero-shot prediction for unseen configurations. Their smallest dataset contained 256 configurations. In contrast, our study operates on much smaller datasets, the largest containing only 30 learning curves, and explicitly models the underlying hierarchical structure. Orthogonal to these are methods transferring optimal hyperparameters from smaller to larger models [20, 47]. In contrast, we fix hyperparameters and zero-shot predict learning curves for larger models.

**Gaussian Processes and Hierarchical Datasets.** Multi-task learning with Gaussian Processes (GPs) was introduced by Bonilla et al. [48], assuming flat structures within the dataset. Hierarchical structures, however, are commonly observed in biological datasets, for example, in modelling gene expression time series [23], organism taxonomies [49], protein families [50], or enzyme classifications [51]. Such datasets often contain multiple layers of hierarchies and a large number of examples [52]. Park and Choi [53] employed hierarchical GPs to model medical data, demonstrating bi-level hierarchy corresponding to clusters of genes and their respective prototypes. Lawrence and Moore [54] introduced hierarchical GP latent variable models for dimensionality reduction, and Hensman et al. [23] proposed a hierarchical kernel designed to capture inherent hierarchies in gene expression datasets. In contrast, we explore hierarchical structures in small datasets with at most 30 examples and a bi-level hierarchy. Building on Ma et al. [24], who introduced latent variable multi-output GPs, capturing hierarchies and task correlations, we use this model for zero-shot learning curve prediction and probabilistic scaling law estimation via Monte Carlo simulation.

**Scaling Laws.** Scaling laws are functional forms that extrapolate the behavior of cost-intensive large language models based on a set of learning curves [4]. These laws provide improved interpretability of neural networks [55, 56, 57], enable effective planning of training sample sizes, and help reduce the associated carbon footprint [36, 58]. Most successful methods in this area are based on empirical, computationally intensive studies [3, 21, 59, 60, 61, 62]. Hägele et al. [59] proposed alternative model training techniques to reduce the computational requirements for obtaining scaling laws. Our work is orthogonal to this approach, as we perform scaling law prediction via Monte Carlo simulation. More related work is provided in Appendix G.

# 3 Zero-Shot Prediction for Probabilistic Scaling Laws

## 3.1 A Brief Introduction to Scaling Laws

In this work, we focus on scaling laws that describe how the validation loss scales with the compute budget $c$. Empirically, this involves generating loss-compute learning curves during training and estimating the parameters of the resulting functional form.

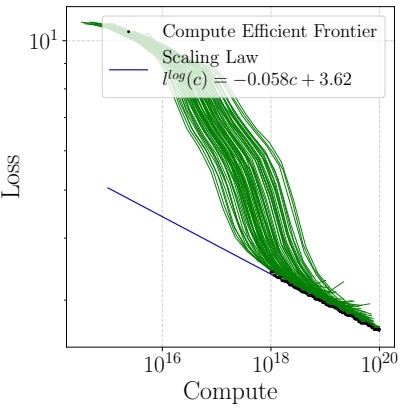

As model and dataset sizes increase, the compute required for a single training iteration also grows, making training progressively more expensive. Let $\mathbf{c} = (c_1, c_2, \ldots, c_M)$ denote an increasing sequence of compute values, with $c_1 < c_2 < \cdots < c_M$. We denote the sequence of loss values achieved by a model of size $n_i$ at these compute values as $\mathbf{l}^{n_i}$. Figure 2 illustrates learning curves $\mathbf{l}^{n_i}$ for nanoGPT models [63] of varying sizes $n_i$ in green and the full set of $Q$ learning curves is given by $\mathcal{D} = \{(\mathbf{c}, \mathbf{l}^{n_i})\}_{i=1}^Q$. Model size is implicitly reflected by the position at which each curve begins on the compute axis: smaller models, requiring less compute per iteration, appear further left.

Figure 2: A set of learning curves from nanoGPT models, each showing loss versus compute for different model sizes. Larger models achieve lower test loss.

While various forms of scaling laws exist, we focus on the $l(c)$ scaling law. Following the approach outlined by Hoffmann et al. [3], this scaling law is obtained by fitting the following functional form to the compute-efficient frontier within a defined compute region of interest: $l(c) = (c/c_0)^{-\gamma}$, with $c_0$ and $\gamma$ being the parameters to be optimized [4, 22]. Figure 2 illustrates the obtained scaling

law (shown in blue) for the nanoGPT dataset in log-log scale. In this representation, the scaling law assumes a linear form. In our study, we aim to estimate the scaling law within a compute-efficient frontier defined for the compute range of $10^{18}$ to $10^{20}$ FLOPs (shown in black). As noted by Hoffmann et al. [3, Figure A5] the specific form of the $l(c)$ scaling law depends on the compute range of interest and the subset of learning curves considered, and can therefore vary accordingly.

## 3.2 Gaussian Process Model for Scaling Law Prediction

In our analysis, we employ the latent variable multi-output Gaussian process (GP) model proposed by Ma et al. [24], hereafter referred to as $\text{Ma}^{\text{GP}}$. In our modelling setup, $t$ indexes tasks and $d$ indexes data instances within each task, forming a bi-level hierarchy. As shown in Figure 1, $t$ may represent embedding parameters, while $d$ corresponds to the number of layers. Thus, models share embedding parameters but differ in layer count, defining distinct model sizes $n_i$ computed from both quantities. The generative model of $\text{Ma}^{\text{GP}}$ is specified as

$$g(\mathbf{x}) \sim \mathcal{GP}(0, k_g(\mathbf{x}, \mathbf{x}')), \quad l_t^d(\mathbf{x}) \sim \mathcal{GP}(g(\mathbf{x}), k_l(\mathbf{x}, \mathbf{x}')),$$
$$y_t^d(\mathbf{x}) = l_t^d(\mathbf{x}, \mathbf{h}_t) + \epsilon_t, \quad \epsilon_t \sim \mathcal{N}(0, \sigma^2), \quad \mathbf{h}_t \sim \mathcal{N}(\mathbf{0}, \mathbf{I}).$$

Each multitask output $y_t^d$ models noisy learning curves with independent and identically distributed Gaussian noise $\epsilon_t$. Each curve $l_t^d(\mathbf{x})$ is drawn from a GP with mean $g(\mathbf{x})$ and covariance $k_l(\mathbf{x}, \mathbf{x}')$. The shared mean function $g(\mathbf{x})$ encodes prior information common to all tasks, with covariance $k_g(\mathbf{x}, \mathbf{x}')$. To model correlations between tasks, a latent variable $\mathbf{h}_t$ is introduced, distributed as $\mathbf{h}_t \sim \mathcal{N}(\mathbf{0}, \mathbf{I})$. The predictive distribution for new values $\mathbf{l}^*$ at inputs $\mathbf{x}^*$ in vector-matrix form can be approximated by $q(\mathbf{l}^*|\mathbf{X}^*) = \int q(\mathbf{l}^*|\mathbf{X}^*, \mathbf{H}) q(\mathbf{H}) d\mathbf{H}$, with $q(\mathbf{l}^*|\mathbf{X}^*, \mathbf{H}) = \mathcal{N}(\mathbf{l}^*|\tilde{\mathbf{m}}_*, \tilde{\mathbf{K}}_*)$. Hence, the posterior predictive distribution over the zero-shot predicted learning curves is given by a multivariate Gaussian distribution with mean $\tilde{\mathbf{m}}_*$ and covariance $\tilde{\mathbf{K}}_*$. This distribution represents the predicted learning curves in our zero-shot setting. As a detailed derivation and discussion of this model is beyond the scope of this paper, the reader is referred to Ma et al. [24].

## 3.3 A Probabilistic Approach to Scaling Laws

The generative scaling law in the log-log domain is given by $l^{\log}(c) \sim \mathcal{N}(\beta_0 + \beta_1 c, \sigma^2)$, with coefficients $\beta_0$ and $\beta_1$. As detailed in Subsection 3.1, its form depends on the available learning curves. Assuming Gaussian prediction error, we estimate $\beta_0$ and $\beta_1$ via Monte Carlo simulation over $R$ runs. Starting from $N$ observed learning curves $\mathcal{D} = \{(\mathbf{c}, \mathbf{l}^{n_i})\}_{i=1}^N$, the $\text{Ma}^{\text{GP}}$ model is trained on $\mathcal{D}$ in each run $r \in R$ to zero-shot predict $M$ missing curves, yielding $\mathcal{D}^* = \{(\mathbf{c}, \mathbf{l}^{m_i*})\}_{i=1}^M$ for unseen model sizes $m_i$. In each run $r$, a scaling law is then fit to the compute-efficient frontier from $\mathcal{D} \cup \mathcal{D}^*$, and final estimates of $\beta_0$ and $\beta_1$ are obtained by averaging over $R$ runs: $\hat{\beta}_0 = 1/R \sum_{r=1}^R p(\beta_0^r \mid \mathcal{D}^*)$ and $\hat{\beta}_1 = 1/R \sum_{r=1}^R p(\beta_1^r \mid \mathcal{D}^*)$ [64].

## 4 Defining Hierarchies in Natural Language Processing Datasets

In the following, we describe the hierarchical structures of each dataset. Hierarchies are exchangeable, as we will show in the experimental section. More details on the models are provided in Appendix B.

**Learning Curves for Scaling Laws.** The nanoGPT$^{\text{large}}$ dataset used for scaling law experiments is shown in Figure 3, with learning curves defined as loss over step size. For the final experiments, these will be converted to learning curves depicting the loss as a function of compute. Each task $t$ corresponds to a row, with data instances $d$ across columns, forming the

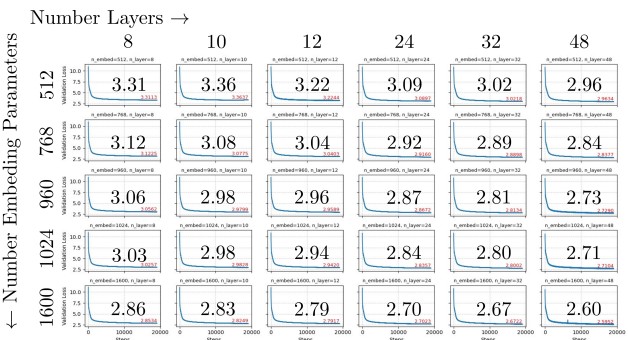

Figure 3: The nanoGPT$^{\text{large}}$ dataset. LCs for different combinations of embedding parameters and layers are shown, with the minimum achieved loss indicated above each curve.

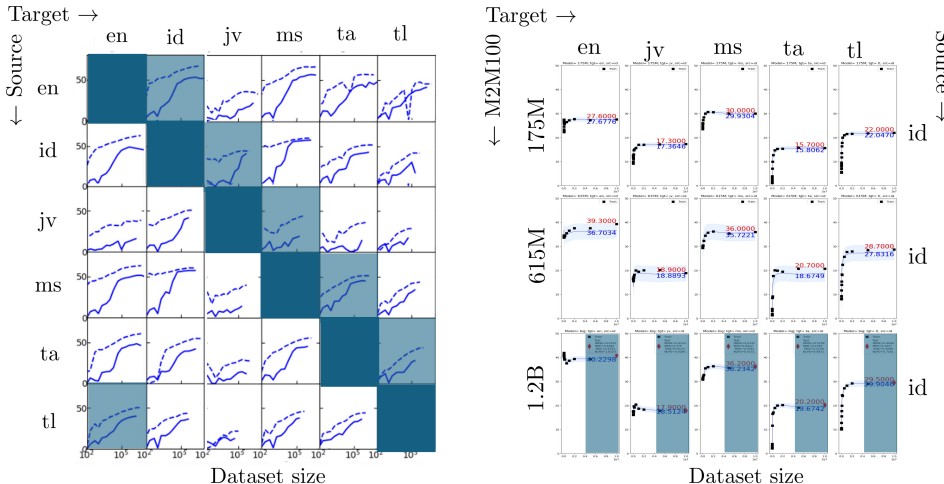

Figure 4: (left) Bilingual learning curve dataset. No data is available on the main diagonal. Light blue squares indicate test-set performance curves. Dashed lines: learning curves from the fine-tuned mBART50 model. Solid lines: learning curves from the Transformer model trained from scratch. (right) Multilingual learning curve dataset. We perform few-shot extrapolation for the M2M100-1.2B model, translating from Indonesian (id) into multiple target languages. The 615M and 175M models correspond to two separate tasks under this assumption. Light blue sections of the curves mark the extrapolated data points.

hierarchical structure illustrated in Figure 1 (left). Here, $t$ indexes embedding parameters and $d$ the number of layers, defining model size $n_i$. Except for combinations 512-8 and 512-10, loss decreases when the number of layers or embedding parameters is increased. (Additional evidence that loss decreases with increasing model size can be found in Kaplan et al. [4] and Henighan et al. [21].) Consequently, this supports the assumption that including correlations among tasks into the modelling process is beneficial for zero-shot prediction. Additionally, we experiment with a subset, referred to as nanoGPT$^{\text{small}}$, containing learning curves for embedding parameter sizes 512, 960, and 1600, and for layer counts 8, 12, 32, and 48.

**Learning Curves for Bilingual Translation.** The bilingual dataset is created by fine-tuning mBART50 and training a Transformer on the EMNLP2021 dataset [65, 66, 67]. Learning curves (Figure 4, left) show translation performance in ChrF [68] over increasing dataset sizes. Each language involved in the dataset introduces distinct linguistic properties, such as word order, morphological complexity, and syntactic structure. The translation performance of a model is not only influenced by the size of the dataset but also by these linguistic characteristics [69, 70, 71]. Prior work shows that factors like typological similarity, dataset size, and source-language pretraining influence performance outcomes [6, 7, 15, 72, 73]. Additionally, in fine-tuning scenarios, model performance has been shown to depend on whether the source language was part of the model's pretraining corpus [74]. Consequently, we structure the dataset hierarchically to improve prediction accuracy. Each task $t$ (row) corresponds to a translation from a specific source language. As a translation into a target language is equally influenced by its linguistic properties, correlations among tasks $d$ correspond to correlations among target languages (Figure 1 (middle)).

**Learning Curves for Multilingual Translation using Architectures of Various Sizes.** Specifically, we investigate extrapolating the learning curve of the largest, most expensive model, assuming learning curves of smaller models are available. Curves represent BLEU performance over increasing dataset sizes, which improves with model size [4]. Each task $t$ corresponds to translation from a source language to target languages $d$ using model size $n_i$ (Figure 1 (right)). Figure 4, right, illustrates translation learning curves from Indonesian into multiple target languages across different model sizes. Leveraging correlations among curves of different model sizes for the same source-target pairs is expected to improve prediction. Zero-shot prediction is not applicable for learning curves obtained after multilingual fine-tuning, since all curves are already available. Therefore, we focus on few-shot prediction for the 1.2B model to estimate expected performance over increasing dataset sizes, based on the performance of smaller models and the available data points of the largest model.

# 5 Experiments

The code for all experiments will be made publicly available. Evaluation metrics are reported as averages over ten independent runs for each model. The MaGP model is trained on z-score-normalized learning curves and logarithmic step or dataset size values, but evaluated in the original domain.

**Common Baseline for all Tasks - Deep Hierarchical Gaussian Processes.** Hensman et al. [23] introduced the Deep Hierarchical Gaussian Process (DHGP) model capable of capturing patterns in data by modelling a two-layer hierarchical relationship, specifically between tasks $t$ and data samples $d$. Extending this framework with an additional hierarchy layer allows the model to identify clusters, i.e., groups of similar tasks $t$ [23, 24]. More information about this model is given in Appendix C. DHGP is particularly suited for discovering clusters in the data, where each cluster can be thought of as a group of tasks sharing similar characteristics (e.g., number of embedding parameters). This is in contrast to MaGP, which assumes that inter-task correlations exist and can be captured through a latent variable. If such correlations are weak or non-existent, we expect the hierarchical structure of DHGP to provide a more appropriate inductive bias than MaGP, allowing it to learn within-cluster trends effectively.

**Additional Baselines for Each Task.** In addition to evaluating DHGP, we include distinct baselines for each task. This differentiation is necessary because the datasets vary in nature, and the corresponding learning curves are structured differently in each setting. A brief introduction to each baseline is given in Appendix D, with more details provided in the following.

## 5.1 Zero-Shot Prediction for Scaling Law Research

The nanoGPT dataset comprises learning curves of loss evaluated over 19,072 validation steps, corresponding to one complete pass through the FineWeb-Edu (10B) dataset [75]. Due to computational constraints, the nanoGPT datasets are subsampled to 11 data points per learning curve, resulting in discretised loss-compute learning curves. We employ the train-test splits Quad and Tri, as illustrated in Figure 5. Additionally, we consider a train-test split containing only the learning curves of the largest five models in the test set, referred to as T1. The largest models in the dataset correspond to the most computationally expensive ones; by predicting their performance, we aim to derive accurate scaling laws at significantly reduced computational cost.

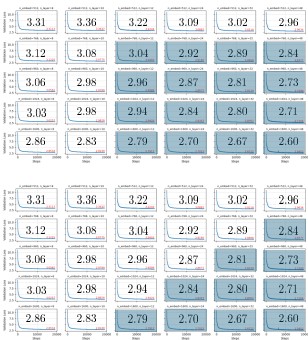

Figure 5: Schematic showing the train-test splits of the nanoGPT[large] dataset: Quad (top) and Tri (bottom). Blue occluded learning curves are the test data.

**Alternative Baseline - Bayesian Neural Networks.** Bayesian Neural Networks (BNNs) have shown promising results in zero-shot learning curve prediction [18]. We compare two variants: BNN (orig), which uses the originally proposed basis functions (vapor-pressure, Hill3, and logarithmic) adapted to model decaying learning curves; and BNN (LC), which employs basis functions better suited to capture decaying learning curves trends, namely the power law, exponential decay, and logarithmic decay functions. The functional forms and illustrations of the used basis-functions are given in Appendix E. Instead of a (conventional) hyperparameter configuration space, we model a configuration space defined by the number of layers and embedding parameters, normalized to the range $[0, 1]$ between the smallest and largest model sizes. Based on previously observed training data, zero-shot predictions are made for novel layer-embedding configurations in the test set.

**Analysis of Zero-Shot Prediction of Learning Curves.** The zero-shot prediction performance of MaGP and its baseline methods is shown in Table 1. Across all metrics, MaGP consistently outperforms the baselines. This result supports our core hypothesis that this NLP dataset exhibits a hierarchical structure that can be effectively leveraged. As expected, reducing the dataset to nanoGPT[small] degrades MaGP's performance, reflecting the increased difficulty due to fewer available learning curves for training. Among the BNN variants, BNN (LC) performs slightly better than BNN (orig), though both show the weakest overall results, likely due to the limited training data. For consistency, all models were evaluated using a fixed sample size of 11 points per learning curve. In comparison, the smallest dataset used in Klein et al. [18] contained 256 configurations, whereas the largest dataset in our study comprises only 29 learning curves, each with 11 data points. This

limitation also explains why we do not report BNN results for T1$^{\text{small}}$ on this dataset. Nevertheless, model performance can be substantially improved with more data points. As shown in Appendix E, Figure E.3, all error metrics improve once the sample size exceeds approximately 100 data points. It should also be noted that the relatively high prediction uncertainty is primarily due to the inherent noise in our learning curve dataset.

Table 1: Zero-shot prediction performance of Ma$^{\text{GP}}$, DHGP, BNN (orig), and BNN (LC) on the nanoGPT$^{\text{large}}$ dataset (Train-Test (TT) splits: Quad, Tri, T1) and zero-shot performance on the nanoGPT$^{\text{small}}$ dataset (TT split: T1$^{\text{small}}$)

| Method | TT | ↓MSE | ↓MAE | ↓MNLPD |
|---|---|---|---|---|
| Ma$^{\text{GP}}$ | Quad | **0.03 ± 0.01** | **0.12 ± 0.02** | **2.58 ± 2.56** |
| DHGP | Quad | 0.07 ± 0.00 | 0.20 ± 0.01 | 3.25 ± 0.24 |
| BNN (LC) | Quad | 10.69 ± 0.43 | 2.82 ± 0.03 | 596.05 ± 23.79 |
| BNN (orig) | Quad | 13.96 ± 0.75 | 3.06 ± 0.05 | 778.95 ± 41.90 |
| Ma$^{\text{GP}}$ | Tri | **0.02 ± 0.01** | **0.10 ± 0.02** | **0.87 ± 0.29** |
| DHGP | Tri | 0.07 ± 0.00 | 0.19 ± 0.00 | 1.85 ± 0.63 |
| BNN (LC) | Tri | 11.02 ± 0.47 | 2.82 ± 0.04 | 612.30 ± 26.22 |
| BNN (orig) | Tri | 13.67 ± 0.82 | 3.03 ± 0.06 | 759.96 ± 45.42 |
| Ma$^{\text{GP}}$ | T1 | **0.04 ± 0.05** | **0.12 ± 0.08** | **0.80 ± 1.54** |
| DHGP | T1 | 0.08 ± 0.00 | 0.18 ± 0.00 | 0.87 ± 0.09 |
| BNN (LC) | T1 | 10.39 ± 0.86 | 2.70 ± 0.07 | 590.54 ± 48.97 |
| BNN (orig) | T1 | 14.05 ± 1.39 | 3.00 ± 0.12 | 798.85 ± 79.19 |
| Ma$^{\text{GP}}$ | T1$^{\text{small}}$ | **0.08 ± 0.00** | 0.21 ± 0.00 | **0.50 ± 0.06** |
| DHGP | T1$^{\text{small}}$ | 0.11 ± 0.00 | 0.21 ± 0.00 | 0.99 ± 0.42 |

A detailed overview of the performance on the last predicted data point, as well as the last three data points in the zero-shot prediction setting, is provided in Appendix H. An error analysis of these points highlights whether the trend of each learning curve was captured accurately. In both cases, Ma$^{\text{GP}}$ significantly outperforms the other methods on the nanoGPT$^{\text{large}}$ dataset and exhibits comparable performance on the nanoGPT$^{\text{small}}$ dataset.

In our initial experiments, we set the number of embedding parameters as the first hierarchy and the number of layers as the second. *But are these hierarchies interchangeable?* To validate our additional hypothesis that hierarchies are indeed exchangeable, we present results for the Ma$^{\text{GP}}$ and DHGP models in Table 2. Here, each number of layers defines a task $t$, while models with the same layer count but different numbers of embedding parameters form the second hierarchy level $d$.

Table 2: Zero-shot prediction performance of Ma$^{\text{GP}}$ and DHGP with exchanged (exch) hierarchies on nanoGPT$^{\text{large}}$ for train-test (TT) splits Quad, Tri, and T1.

| Method | TT | ↓MSE | ↓MAE | ↓MNLPD |
|---|---|---|---|---|
| Ma$^{\text{GP}}$ exch | Quad | **0.04 ± 0.04** | **0.11 ± 0.07** | 1.82 ± 2.23 |
| DHGP exch | Quad | 0.11 ± 0.00 | 0.26 ± 0.01 | **0.34 ± 0.02** |
| Ma$^{\text{GP}}$ exch | Tri | **0.02 ± 0.01** | **0.09 ± 0.03** | **0.37 ± 0.79** |
| DHGP exch | Tri | 0.13 ± 0.05 | 0.29 ± 0.06 | 0.48 ± 0.16 |
| Ma$^{\text{GP}}$ exch | T1 | **0.01 ± 0.02** | **0.08 ± 0.04** | **−0.88 ± 0.58** |
| DHGP exch | T1 | 0.16 ± 0.02 | 0.34 ± 0.02 | 1.18 ± 0.43 |

The results show that Ma$^{\text{GP}}$ still outperforms the baseline when the hierarchy is exchanged, supporting our hypothesis. Subsequent experiments and analyses continue with the original hierarchical assumption illustrated in Figure 1.

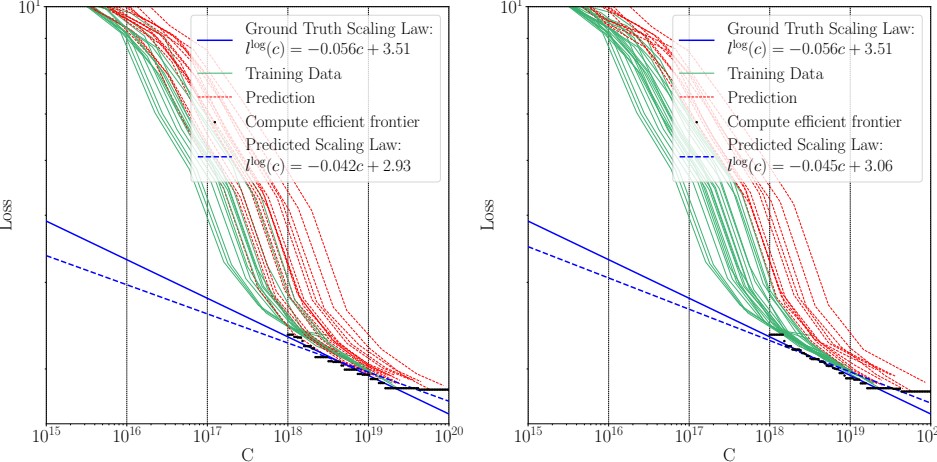

Figure 6: Predicted learning curves (red) using training curves (green) for Quad (left) and Tri (right) training-test splits in a single run. Dashed blue lines show the predicted scaling laws; solid blue lines show the ground truth scaling law.

**Scaling Law Analysis.** Figure 6 shows the Quad and Tri train-test splits and the predicted scaling law derived from the predicted learning curves $\mathcal{D}^*$ and the available training data $\mathcal{D}$ for a single run. We consider a compute range between $10^{18}$ and $10^{20}$ FLOPs. Note

Table 3: Predicted scaling laws and Area between Curves (AbC) to the ground truth law $(-0.056c + 3.51)$, averaged over 10 runs for different Train-Test (TT) splits. Reported (compute) costs for obtaining the training data are given in PetaFLOPs (full nanoGPT$^{\text{large}}$: $6.081 \times 10^5$ PetaFLOPs).

| TT | Predicted Scaling Law | ↓AbC | ↓Cost |
|------|-----------------------|------|-------|
| Quad | $(-0.043 \pm 0.002)c + (2.957 \pm 0.074)$ | $0.521 \pm 0.069$ | $1.28 \cdot 10^5$ |
| Tri | $(-0.052 \pm 0.005)c + (3.352 \pm 0.202)$ | $0.160 \pm 0.172$ | $2.06 \cdot 10^5$ |
| T1 | $(-0.059 \pm 0.006)c + (3.636 \pm 0.245)$ | $0.111 \pm 0.222$ | $5.15 \cdot 10^5$ |

that due to using a subset of the full dataset, the ground-truth scaling law for nanoGPT$^{\text{large}}$ differs from that in Figure 2, which was obtained from a larger dataset of 90 learning curves. Recent scaling law studies typically estimate the blue scaling law using the full dataset, $\mathcal{D} \cup \mathcal{D}^*$, by exhaustively training all models rather than leveraging zero-shot prediction to reduce computational cost. These results serve as the 'ground truth' baseline. The compute-efficient frontier appears discretised rather than smooth because each learning curve contains only 11 data points. Using Monte Carlo averaging over 10 runs, the resulting predicted scaling laws are summarized in Table 3. Additionally, to quantify the deviation between the predicted and ground-truth scaling laws, we compute the Area between Curves (AbC) metric, where AbC $\in (0, \infty)$. The concept is illustrated in Appendix F. For consistency, AbC is evaluated over the compute range $10^{13}$-$10^{23}$ FLOPs, with smaller values indicating closer alignment between predicted and ground-truth scaling laws. For each train-test split (TT), the slope coefficient is estimated with higher certainty than the intercept, which shows greater variability. As expected, increasing the amount of training data reduces the mean AbC but also increases its uncertainty. While the Quad split is the most compute-efficient in terms of training data acquisition, it yields the highest AbC. In contrast, the Tri and T1 splits achieve lower AbC values but at the cost of higher predictive uncertainty. This motivates the question of whether a more informative train-test split strategy can be designed. To explore this, and inspired by active learning, we investigate the effect of different query strategies on AbC performance [76, 77, 78].

**Analysis of Various Query Strategies.** We begin with an initial training set of learning curves $\mathcal{D}$ from the nanoGPT$^{\text{large}}$ dataset, consisting of one curve per task $t$ corresponding to the smallest layer size (i.e., the first column in Figure 3). At each iteration, one additional learning curve is queried according to a predefined query strategy. After each query, the Ma$^{\text{GP}}$ model is retrained on the existing training set $\mathcal{D}$ together with the newly queried curve. Subsequently, the test set curves $\mathcal{D}^*$ are predicted (these are the remaining curves in the nanoGPT$^{\text{large}}$ dataset). Predictions are averaged over 10 runs to capture variability. Following each prediction step, a scaling law is fitted, and its performance is assessed using the AbC metric.

We evaluate four query strategies. The first, *Largest First*, selects the learning curve corresponding to the largest model from the test set; the second, *Smallest First*, selects the learning curve corresponding to the smallest model; the third, *Active Learning*, queries the most uncertain learning curve based on predictive variance; and finally, we use *Random Order* for a random choice of the next learning curve to be queried. For the *Active Learning* strategy, the predictive mean $\overline{\mathbf{l}^{m_i*}}$ of each learning curve is obtained by averaging over 10 runs. The variance at each sample point is then computed as $\text{var} = 1/10 \sum_{i=1}^{10} \left( \mathbf{l}^{m_i*} - \overline{\mathbf{l}^{m_i*}} \right)^2$. Finally, the mean variance per learning curve, denoted as mvar, is calculated by averaging over its 11 sampled points: $\text{mvar} = 1/11 \sum_{j=1}^{11} \text{var}_j$.

The results of these query strategies, along with a cost analysis over the number of queries, are presented in Figure 7. Querying according to *Active Learning* consistently results in more certain AbC values across runs. Moreover, these values are either below the other query strategies (e.g. *Smallest First*) or comparably high (e.g. *Largest First*). In contrast, random querying introduces a substantial variability and uncertainty into the predictions made by Ma$^{\text{GP}}$. Although the *Largest First* strategy achieves AbC values similar to *Active Learning*, it incurs higher computational cost and larger standard deviations. Overall, averaged over 10 runs, *Active Learning* produces the most confident learning curve predictions and the most reliable predicted scaling laws. For all query counts, the AbC from *Active Learning* remains consistently below (or comparable to) that of the initial training set (query number 0), a property not shared by the other strategies. The order of queried LCs for each scenario is illustrated in Appendix J. Additionally, Appendix I, Figure I.1 presents AbC values as a function of computational cost.

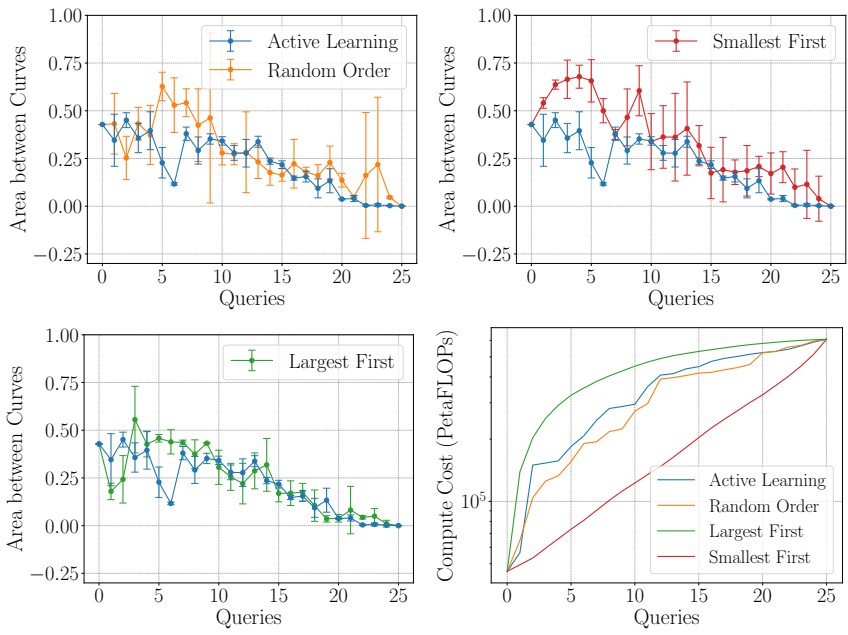

Figure 7: AbC over queries using various query strategies and a cost analysis.

## 5.2 Zero-Shot Performance Prediction for Bilingual Translation

To further support our core hypothesis that learning curve datasets exhibit hierarchical structures, we conduct additional zero-shot prediction experiments using bilingual translation data. The corresponding train-test split is illustrated in Figure 4 (left). While the figure shows learning curves measured in ChrF as a function of dataset size, we also evaluate zero-shot prediction performance using BLEU scores. Unlike the nanoGPT dataset, this dataset is considerably noisier and more variable, with each learning curve containing between 11 and 17 points, depending on the available data for each source-target language pair.

**Alternative Baseline.** Assuming correlations among learning curves sharing the same source $s'$ or target $t'$ language, we define a naive baseline (NBL) as the average curve over these respective sets for a language pair $s't'$ of interest, given by: $1/2 \left( 1/S \sum_{s=1}^{S} \mathbf{l}_{st'} + 1/T \sum_{t=1}^{T} \mathbf{l}_{s't} \right)$, for $s' \neq s$ and $t' \neq t$, where $S$ and $T$ denote the number of available source and target languages, respectively.

The results are shown in Figure 8. BLEU and ChrF performances are not directly comparable due to differing scales. Ma$^{GP}$ exhibits higher predictive uncertainty when forecasting BLEU learning curves but consistently outperforms all baselines in terms of mean RMSE. NBL shows substantial variability, indicating that simple averaging over source and target language curves fails to capture meaningful correlations. While DHGP improves over NBL, it still displays heightened predictive uncertainty. These findings confirm our core hypothesis that modeling bilingual translation tasks with two-layer hierarchical structures allows the model to leverage correlations among tasks, enhancing prediction accuracy. Further analysis is provided in Appendix K and Appendix L, and results for exchanged hierarchies supporting our additional hypothesis are in Appendix M.

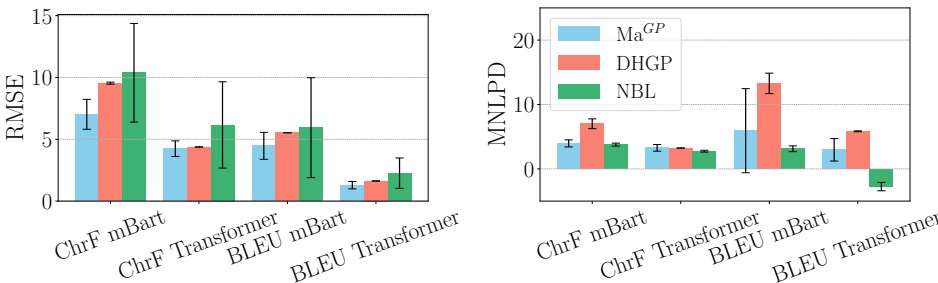

Figure 8: Zero-shot performance prediction on the bilingual dataset.

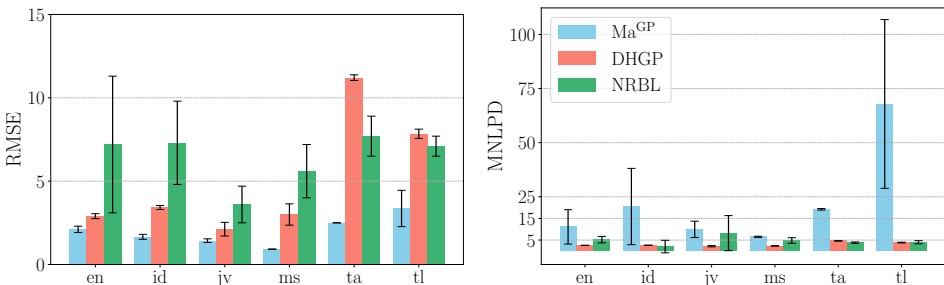

Figure 9: Few-shot performance prediction on the multilingual dataset, extrapolating the last nine data points of each learning curve for the M2M100 1.2B model. The extrapolated learning curves show performance over dataset size for translations into a common **target** language.

## 5.3 Few-Shot Performance Prediction for Architecture Design

To provide a final experimental setting for validating our core hypothesis, we conduct few-shot performance prediction of learning curves across various model architectures in multilingual translation. We assume that learning curves from already fine-tuned M2M100 models of sizes 175M and 615M are available. Using these, we extrapolate the final nine missing points of the 1.2B model's learning curves, corresponding to the largest dataset sizes. This allows us to assess whether collecting additional data for translation into specific target languages or from particular source languages is justified, based on the predicted performance. The extrapolation leverages both the learning curves of the smaller models and the initial performance values of the largest model.

**Alternative Baseline.** We define a naive regression baseline (NRBL) as a non-linear regression model trained on the available initial training data points of the 1.2B model, together with the complete learning curves of the 175M and 615M models. This model fits several functional forms to the training data, including a vapor-pressure function, the MMF function, a power law function, and a logarithmic function. These functions are given in Appendix O. The reported performance is the average across these fitted model.

Figure 9 shows the extrapolation performance when predicting into a common target language. Using Ma$^{\text{GP}}$, we consistently outperform the baseline models in terms of RMSE, despite higher predictive uncertainty. The highest uncertainty is observed for learning curves predicting performance over dataset size for Tagalog (*tl*) as a target language. Nevertheless, mean predictions across all languages and translation scenarios achieve RMSE values superior to the baselines. These results confirm that incorporating hierarchical assumptions and leveraging correlations among tasks enhances prediction performance. Further analysis is provided in Appendix N. The hierarchical structures in this setting are exchangeable as demonstrated in Appendix N. This aligns with the previous results and confirms our additional hypothesis.

## 6 Conclusion

We demonstrate that framing learning curve prediction tasks in natural language processing as multi-task problems, in which each task is organized according to a two-layer hierarchical structure and correlations among tasks are leveraged, leads to substantial improvements in zero-shot prediction performance. Through experiments across three small-scale datasets, we provide empirical evidence supporting our core hypothesis that learning curve datasets in NLP inherently exhibit a bi-level hierarchical structure. Furthermore, our results show that these hierarchies are exchangeable, highlighting the robustness and flexibility of the hierarchical modeling approach in capturing underlying task relationships and enabling accurate prediction even when the ordering of hierarchical levels is altered.

Building on the Ma$^{\text{GP}}$ framework, we further demonstrate that probabilistic scaling laws can be derived through Monte Carlo simulation, supporting our final hypothesis. By employing an active learning strategy, we observe a notable reduction in model uncertainty on average, and the resulting predicted scaling laws closely align with the ground truth, highlighting the effectiveness of combining hierarchical modeling with principled uncertainty quantification.

Further discussion and additional details on computational complexity are provided in Appendix Q and Appendix P, respectively. An impact statement and limitations are given in Appendix A.

## Acknowledgment

We would like to thank the anonymous reviewers for their constructive and insightful feedback on the initial version of this paper. Their thoughtful comments and engagement with our work have led to valuable improvements, which we believe have further strengthened the quality and contribution of this study.

This research was supported by The University of Melbourne's Research Computing Services and the Petascale Campus Initiative. MH was supported by the Australian Research Council (ARC) through grant DP230102775.

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

# A  Impact Statement and Limitations

**Impact Statement.**   Considering the performance achieved with limited computational resources and a small number of samples per learning curve, these results demonstrate the superiority and strong potential of Ma$^{\text{GP}}$ for zero-shot prediction.

In the context of scaling laws, the proposed approach offers a cost-efficient strategy for determining scaling behaviors while providing valuable uncertainty estimates. When combined with active learning, it enables an effective trade-off between uncertainty reduction and computational cost.

For bilingual translation tasks, assuming performance curves for certain datasets are already available, the proposed framework can save expensive training and testing times via zero-shot performance prediction of bilingual translation performances for other language-pairs of interest.

The implications for architecture design are substantial. In particular, Ma$^{\text{GP}}$, within our experimental setting, can support informed decision-making regarding dataset sizing to meet specific performance objectives when deploying larger model architectures.

**Limitations.**   Due to computational constraints, we were limited to 11 sample points per learning curves for the scaling law experiments, and a five times six grid of learning curves. While our results already demonstrate the advantages of Ma$^{\text{GP}}$, we suspect that more sampling points and larger grid could show additional interesting insights using this framework.

Another current limitation of Ma$^{\text{GP}}$ is the interpolation across the architectural grid. Specifically, our dataset is structured on a fixed grid defined by the number of layers and embedding parameters. As a result, the model in its current form cannot interpolate to unseen values of embedding parameters. Interpolation across unseen numbers of layers is technically feasible, but it would yield an averaged estimate between the nearest known configurations, which may not reflect true model behavior. We see this as a promising direction for future work and plan to investigate more flexible representations to address this limitation (e.g. w/ meta parameters).

The variance-based active learning strategy explored in our work focuses on reducing the uncertainty, independent of associated compute costs. Note that there exist many different ways how the cost-function in the active learning process could be structured, but this is outside the scope of this paper. We see the design of cost-terms that consider trade-offs like compute-vs-uncertainty-reduction as an interesting direction of future work.

# B  Newly Created Datasets and their Specifications

## B.1  Scaling Laws

**nanoGPT.**   We train all nanoGPT models [63] on the 10B token split of the FineWeb-Edu dataset [75] using four NVIDIA A100 (80GB) GPUs. A learning rate of $6 \cdot 10^{-4}$ is applied with linear warm-up and cosine decay scheduling, a minimum learning rate of $5 \cdot 10^{-4}$, and a total batch size of 524,288.

NanoGPT was introduced in 2022 and is a minimal, educational implementation of GPT (Generative Pre-trained Transformer) [79]. The model implements the standard GPT architecture with, multi-head self-attention mechanisms, feed-forward networks, layer normalization, positional embeddings and Transformer decoder blocks. This architecture is representative of the Transformer-based models widely used in state-of-the-art natural language processing. Since its introduction, it has been used to conduct research on novel model architectures [80], on topographic models [81] and on numerical reasoning [82] (among others).

## B.2  Bilingual Translation

**mBART.**   mBART50 is a multilingual sequence-to-sequence model trained with a denoising objective over 50 languages, using special language tokens to indicate the desired target language for generation. At decoding time, a token ID must be specified to direct the model to translate into the appropriate target language. Following the approach of Lee et al. [74], we select related languages for tokenisation in cases involving unseen languages. Based on syntactic and phylogenetic proximity, for example, the *id* tokenizer is applied for both *jv* and *ms* [83, 84].

To generate a new dataset of learning curves for bilingual translation, we fine-tune the mBART50 model using the HuggingFace implementation [85]. Fine-tuning is performed on NVIDIA A100 GPUs for one epoch [86], with a learning rate of $5 \cdot 10^{-5}$, a dropout rate of $0.1$, maximum sequence lengths of 200 for both source and target texts, and a batch size of 10. For decoding, beam search is employed with a beam size of 5. For each language direction, the encoder-decoder parameters are initialized from the pretrained mBART50 model's corresponding encoder and decoder components.

**Transformer.** To create a new dataset of learning curves for bilingual translation using Transformer models, we use the implementation provided by fairseq [87] and train on NVIDIA A100 GPUs. Depending on the size of the training dataset, two model configurations are employed. For datasets containing fewer than 10k parallel sentences, the model architecture comprises 3 encoder and 3 decoder layers, each with embedding dimensions of 512 and 2 attention heads. For dataset sizes equal to or exceeding 10k parallel sentences, the architecture consists of 6 encoder and 6 decoder layers, with an embedding dimension of 256 and 2 attention heads [74].

Training is initiated with a learning rate of $1 \cdot 10^{-3}$, a weight decay of $1 \cdot 10^{-4}$, a dropout rate of $0.4$, and a batch size of 32. Early stopping is applied based on the validation loss. Input sequences are tokenised into subword units using SentencePiece [88]. For decoding, beam search with a beam size of 5 is employed.

**Train-Test Dataset.** We use the small track dataset from the EMNLP2021 shared task on large-scale multilingual machine translation[2] for fine-tuning mBART50 [74] and training the Transformer model [66]. This dataset consists of sentences extracted from the OPUS corpus[3], which encompasses data of varying quality across multiple domains. The languages contained in this dataset are Tamil (*ta*), Tagalog (*tl*), Malay (*ms*), Javanese (*jv*), Indonesian (*id*) and English (*en*). The number of sentences available for each translation pair varies, reflecting a realistic scenario with learning curves of variable lengths for model evaluation. Consistent with the original shared task, we employ the Flores101 dataset [89] as the test set and report BLEU [90] and ChrF scores.

The total number of sentences available for each language pair are given in Table B.1. Our learning curves exclusively represent low-resource scenarios for the language pairs id-jv / jv-id, id-ta / ta-id, jv-ms / ms-jv, jv-ta / ta-jv, jv-tl / tl-jv, ms-ta / ta-ms, and ta-tl / tl-ta (denoted as LR in Table B.1). Among these, the language pairs jv-ms / ms-jv, jv-ta / ta-jv, and ms-ta / ta-ms will represent extremely low-resource scenarios (eLR). Consequently, the dataset provides 14 out of 30 language pairs that reflect real-world conditions of limited bilingual corpora availability, meaning no sufficient additional data is available to create a learning curve data point indicative of a high-resource setting for these pairs. It is important to note that all learning curves data points correspond to extremely low-resource conditions for dataset sizes below $0.5$ million sentences, and to low-resource conditions for sizes below 1 million sentences. Only 16 out of the 30 language pairs will have performance data points

---

[2]`https://www.statmt.org/wmt21/large-scale-multilingual-translation-task.html`
[3]`https://opus.nlpl.eu/`

Table B.1: Total number of available sentences per language pair, sorted from largest to smallest.

| Language Pair | No. Sentences |
|---|---|
| en-id / id-en | 54,075,906 |
| en-tl / tl-en | 13,612,416 |
| en-ms / ms-en | 13,437,738 |
| en-jv / jv-en | 3,044,926 |
| id-tl / tl-id | 2,743,314 |
| id-ms / ms-id | 2,000,000 |
| en-ta / ta-en | 2,115,938 |
| ms-tl / tl-ms | 1,358,493 |
| jv-tl / tl-jv | 817,149 |
| id-jv / jv-id (LR) | 780,125 |
| ta-tl / tl-ta (LR) | 563,342 |
| ms-ta / ta-ms (LR) | 372,631 |
| jv-ms / ms-jv (eLR) | 434,714 |
| id-ta / ta-id (eLR) | 500,000 |
| jv-ta / ta-jv (eLR) | 66,000 |

available for high-resource bilingual machine translation settings, i.e., dataset sizes of above 1 million sentences.

A low resource-setting can occur if either the language or the domain of interest are low-resourced [91]. LRL could be threatened or endangered languages for which little spoken/written (linguistic) resources or datasets, or number of native speakers/experts are available [91, 92, 93].

### B.3 Multilingual Translation

In multilingual neural machine translation, the objective is to train a single model capable of translating between multiple language pairs [94]. Such multilingual models offer practical advantages, as they reduce the total number of models and overall model parameters required, thereby simplifying deployment (compared to bilingual machine translation). Additionally, multilingual NMT models promote transfer learning: when low-resource language pairs are trained alongside high-resource pairs, the shared representations and parameter sharing can lead to notable improvements in translation quality for the low-resource languages [95, 96, 97].

Traditionally, neural machine translation systems have been predominantly English-centric, with most training data involving English as either source or target language. However, this does not reflect the broader translation needs of global users [94]. To address this limitation, the M2M100 model was developed.

**M2M100.** This model is capable of translating directly between any pair of 100 languages without relying on English as an intermediate [94], supporting 9,900 translation directions. To pre-train M2M100, a large-scale dataset was constructed through extensive mining of parallel corpora, encompassing a wide range of language pairs beyond those involving English. This dataset enables the model to learn direct translations between numerous language combinations.

We construct a new dataset of learning curves derived from the 175M, 615M, and 1.2B parameter M2M-100 models, fine-tuned on the EMNLP2021 dataset. M2M100 models are built using the Transformer architecture [66] and implemented via fairseq [87]. We employ the Adam optimizer with parameters $\beta_1 = 0.90$, $\beta_2 = 0.98$, and a weight decay of $0.0001$. The training objective is the label-smoothed cross-entropy criterion with label smoothing of $0.1$. The initial learning rate is set to $0.0003$, scheduled by the inverse square root learning rate scheduler with 2,500 warm-up steps. The batch size (number of tokens) is $4096 \times 32$ for the 175M model, and $2048 \times 64$ for both the 615M and 1.2B models. Additional architectural details are provided in Table B.2.

**Train-Test Dataset.** We use the same train-test dataset as in Appendix B.2.

Table B.2: M2M architecture configurations.

| **Model Size** | 175M | 615M | 1.2B |
|---|---|---|---|
| Vocabulary Size | 256K | 256K | 128K |
| Word Representation Size | 512 | 1,024 | 1,024 |
| Feed-forward Layer Dimension | 2,048 | 4,096 | 8,192 |
| Encoder/Decoder Layers | 6 | 12 | 24 |
| Attention Heads | 16 | 16 | 16 |
| Dropout Rate | 0.1 | 0.1 | 0.1 |
| Layer Dropout Rate | 0.05 | 0.05 | 0.05 |

## C  The Generative Model of DHGP

Ma$^{\text{GP}}$ introduces an additional latent variable $\mathbf{h}$ to model correlations among tasks $t$, allowing it to capture cross-task dependencies. In contrast, DHGP does not rely on this shared latent representation. Instead, it leverages a hierarchical modeling approach defined as

$$f(\mathbf{x}) \sim \mathcal{GP}(0, k_f(\mathbf{x}, \mathbf{x}'))$$
$$g(\mathbf{x}) \sim \mathcal{GP}(f(\mathbf{x}), k_g(\mathbf{x}, \mathbf{x}'))$$
$$l(\mathbf{x}) \sim \mathcal{GP}(g(\mathbf{x}), k_l(\mathbf{x}, \mathbf{x}'))$$
$$y(\mathbf{x}) = l(\mathbf{x}) + \epsilon, \quad \epsilon \sim \mathcal{N}(0, \sigma^2).$$

Following and inspired by the approach of Hensman et al. [23], we interpret $f$ as a general function capturing global learning curve trends across tasks. The next level, $g$, captures variations due to a coarser factor, i.e., the number of embedding dimensions. At the lowest level, $l$ models individual learning curves, which vary with the number of layers.

## D  The Reasoning Behind Additional Baselines for Each Experiment

For the nanoGPT$^{\text{large}}$ dataset, we have access to a sufficient number of learning curves and data points, making a Bayesian Neural Network (BNN) a valid and effective baseline. This choice is further motivated by prior work, such as Klein et al. [18], where BNNs were successfully applied for learning curve extrapolation and zero-shot prediction.

In contrast, applying a BNN in the bilingual translation setting is more challenging. While we explored modeling the learning curves as a function of source and target language pairs, the BNN struggled to generalize and capture the upward trend in learning curves. Also note, we only have 25 learning curves for training, and each source and target language is only 5 times represented, which given the noisy nature of this dataset (see main paper, Figure 4, left) is not enough training data to perform well.

Given the assumption that source and target languages may share structural similarities, we instead opted for a simpler and more robust baseline: averaging available learning curves across source and target languages. This approach offers a language-agnostic approximation of performance trends.

The final experiment involves an extrapolation task, where we predict learning curves from a single source (or target) language into five different target (or source) languages. Unlike the bilingual translation setup, the training grid in this scenario is defined by three distinct model sizes (see main paper, Figure 4, right). The dataset for this experiment contains only 15 learning curves, which is (again) insufficient for effectively training a BNN to generalize. Hence, a regression-based baseline, such as polynomial or power-law fitting, is a more suitable and reliable choice.

## E  More Information on Bayesian Neural Networks

BNN LC uses the following basis functions

$$f_{\exp}(x) = a \cdot e^{-bx} + c, \quad f_{\text{pl}}(x) = a \cdot \left(x + 10^{-5}\right)^{-b} + c, \quad f_{\log}(x) = a - b \cdot \log(x + 10^{-5}) + c,$$

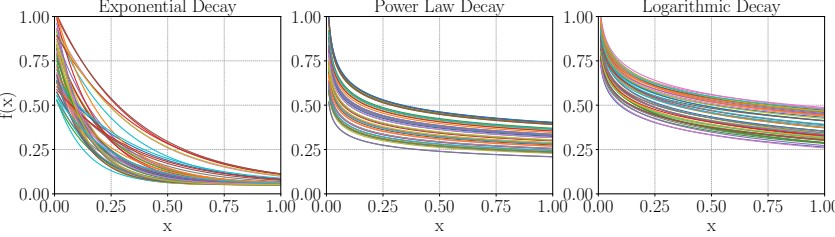

Figure E.1: Samples of basis functions used for BNN LC.

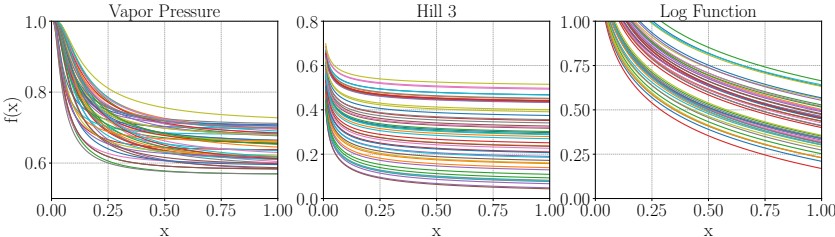

Figure E.2: Samples of basis functions used for BNN orig.

with samples given in Figure E.1.

We adjusted the originally used basis functions, designed for the extrapolation of increasing learning curves to

$$f_{\text{vapor}}(x) = \exp\left(-\exp\left(-a - \frac{b}{x + 10^{-5}} - c \cdot \log(x + 10^{-5})\right)\right),$$

$$f_{\text{log}}(x) = 1 - \frac{c + a \cdot \log(bx + 10^{-10})}{10},$$

$$f_{\text{hill3}}(x) = 1 - a \cdot \left(\frac{1}{\left(\frac{c}{x+10^{-5}}\right)^b + 1}\right),$$

with samples given in Figure E.2.

For the experiments in the main paper, due to computational constraints, each nanoGPT learning curve is subsampled to 11 data points. These 11 data points are equally distributed in log-step size space to be fit using Gaussian Processes. For a fair comparison, we keep this transformation for the BNN model. However, changing the sample size to be equally spaced in normal domain leads to increased performance in terms of RMSE of to below 2.0 as demonstrated in Figure E.3.

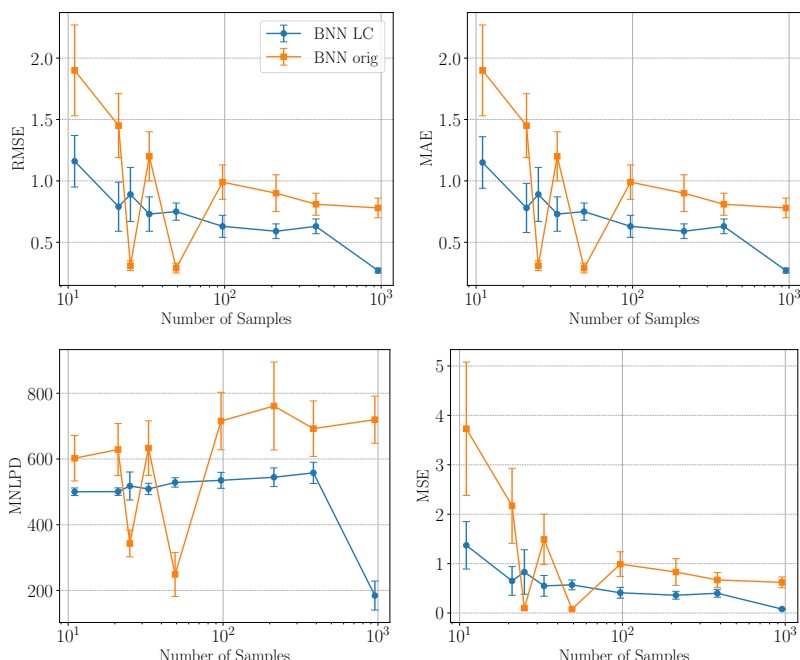

Figure E.3: Zero-Shot BNN performance for increasing sample sizes per learning curves.

# F   More Information on the Introduced Metric: Area between Curves

The Area between Curves (AbC) is illustrated in Figure F.1. In order to apply this measure, one needs to define an AbC range of interest. In this case, it is $10^{13}$ to $10^{23}$ FLOPS (shaded area).

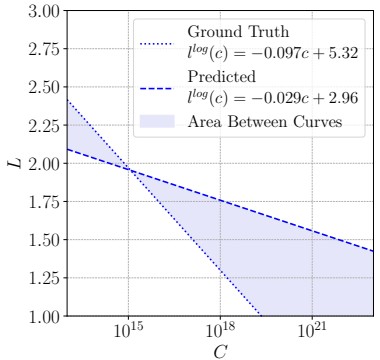

# G   Extended Literature

## G.1   Scaling Laws

The recent surge in scaling law research offers several benefits, including improved interpretability of neural networks [55, 56, 57], more effective training dataset size acquisition, and a reduced carbon footprint [36, 58]. While theoretical approaches have been developed to facilitate the prediction of scaling laws for only a limited number of architectures [55, 98, 99], the majority of successful methods for predicting scaling behavior across a wide range of networks are based on empirical studies [3, 4, 21, 57, 58, 59, 60, 61, 62].

Figure F.1: Area between Curves (AbC) for the compute range of interest being $10^{13}$ to $10^{23}$ FLOPS (shaded area). The ground truth scaling law is compared to an example of a predicted LC.

Scaling laws depict a functional form $f(x)$ that characterizes the development of a performance measure, such as validation loss $L$, relative to a scale, which may refer to compute $C$, the number of samples in the dataset $D$, or the number of parameters in the model $N$. In an empirical approach, LCs are generated, and the parameters of the functional form $f(x) = \beta x^a + c$ are then estimated, for $x$ being the scale of interest, $c$ denoting the irreducible loss [21, 59] and $f(\cdot)$ being the functional form of its performance measure [100]. As shown by Hoffmann et al. [3, Figure A5], depending on the compute range of interest, the slope of $L(C)$ in the log-log domain will vary.

Scaling laws operate under the assumption of an ideal scenario where increasing dataset size corresponds to having consistently same-quality data available. However, as noted in Muennighoff et al. [101], this is not the reality for most languages, where available data is limited, leading to data-constrained large language models (LLMs). Even for English, high-quality data is predicted to be exhausted by 2024 according to the Chinchilla scaling laws, given the current trend of training ever-larger models [102]. In data-limited regimes, strategies such as repeating data can alter scaling law behavior. Moreover, as mentioned in Kaplan et al. [4], scaling law trends must eventually level off, a phenomenon that a straightforward power-law trend cannot predict. In certain cases, such as discriminative models like image classifiers, clear scaling laws do not emerge, and performance may plateau even as dataset or model size increases [103].

## G.2   Active Learning

Active learning (AL) enables a machine learning model to achieve better accuracy with fewer training instances, when the model chooses the training data to learn from. It also leads to significant cost-reductions in scenarios, where labeled data is expensive to obtain [77], i.e., perfectly suited for scaling law research or expensive bilingual or multilingual translation performances. AL is a widely used method in areas such as recommender systems [104], natural language processing [105], medical imaging [106] or image processing [107].

AL strategies query either single data points [108] or batch data points [109]. When acquisition functions are based on a model's uncertainty, they are called model-based [110]. One goal of these functions is to minimize the expected predictive loss of the model [110]. In our scenario we assume 10 runs, i.e., a committee of $10$ $\text{Ma}^{\text{GP}}$ models, whose predictive uncertainty we aim to reduce. We define the maximum disagreement of the models by the largest average (per sample point) variance per learning curve.

# H  Zero-Shot Performance Prediction for Probabilistic Scaling Laws

We are particularly interested in the prediction performance at the final data points of each learning curve, as these indicate the values to which the zero-shot predicted learning curves are expected to converge. An error analysis of these points highlights whether the overall trend of each learning curve has been correctly captured. The results of this analysis are reported in Table H.1 and Table H.2. On the nanoGPT$^{\text{large}}$ dataset, Ma$^{\text{GP}}$ achieves a substantial improvement over all competing methods, while on the nanoGPT$^{\text{small}}$ dataset it maintains performance comparable to the best baselines.

Table H.1: Zero-shot prediction performance on the **last extrapolated point** using Ma$^{\text{GP}}$, DHGP, BNN (orig), and BNN (LC). Results are shown for the nanoGPT$^{\text{large}}$ train-test (TT) splits Quad, Tri, and T1, and for T1$^{\text{small}}$ on the nanoGPT$^{\text{small}}$ dataset.
($*$: Exploding MNLPD values caused by excessive uncertainty and limited training data.)

| Method | TT | ↓MSE | ↓MAE | ↓MNLPD |
|---|---|---|---|---|
| Ma$^{\text{GP}}$ | Quad | **0.01 ± 0.01** | **0.07 ± 0.03** | **−0.93 ± 1.43** |
| DHGP | Quad | 0.03 ± 0.00 | 0.17 ± 0.00 | 0.51 ± 0.08 |
| BNN (LC) | Quad | 14.44 ± 2.21 | 3.79 ± 0.29 | $*$ |
| BNN (orig) | Quad | 23.79 ± 3.58 | 4.86 ± 0.38 | $*$ |
| Ma$^{\text{GP}}$ | Tri | **0.00 ± 0.00** | **0.06 ± 0.02** | **−1.27 ± 0.21** |
| DHGP | Tri | 0.02 ± 0.00 | 0.15 ± 0.00 | −0.22 ± 0.13 |
| BNN (LC) | Tri | 15.26 ± 3.22 | 3.88 ± 0.41 | $*$ |
| BNN (orig) | Tri | 22.32 ± 3.18 | 4.71 ± 0.34 | $*$ |
| Ma$^{\text{GP}}$ | T1 | **0.00 ± 0.01** | **0.05 ± 0.05** | **−1.08 ± 0.92** |
| DHGP | T1 | 0.01 ± 0.00 | 0.12 ± 0.00 | −0.67 ± 0.02 |
| BNN (LC) | T1 | 15.52 ± 3.42 | 3.92 ± 0.40 | 434.26 ± 32.64 |
| BNN (orig) | T1 | 24.22 ± 5.22 | 4.90 ± 0.52 | 532.18 ± 86.57 |
| Ma$^{\text{GP}}$ | T1 small | 0.05 ± 0.00 | 0.22 ± 0.00 | −0.03 ± 0.01 |
| DHGP | T1 small | **0.04 ± 0.01** | **0.19 ± 0.00** | **−0.23 ± 0.04** |

Table H.2: Zero-shot prediction performance on the last **three extrapolated points** using Ma$^{\text{GP}}$, DHGP, BNN (orig), and BNN (LC). Results are shown for the nanoGPT$^{\text{large}}$ train-test (TT) splits Quad, Tri, and T1, and for T1$^{\text{small}}$ on the nanoGPT$^{\text{small}}$ dataset.
($*$: Exploding MNLPD values caused by excessive uncertainty and limited training data.)

| Method | TT | ↓MSE | ↓MAE | ↓MNLPD |
|---|---|---|---|---|
| Ma$^{\text{GP}}$ | Quad | **0.00 ± 0.01** | **0.06 ± 0.03** | **−1.11 ± 1.12** |
| DHGP | Quad | 0.03 ± 0.01 | 0.15 ± 0.00 | 0.14 ± 0.07 |
| BNN (LC) | Quad | 14.26 ± 2.16 | 3.76 ± 0.28 | 480.69 ± 9.49 |
| BNN (orig) | Quad | 23.85 ± 3.36 | 4.86 ± 0.35 | 582.48 ± 49.97 |
| Ma$^{\text{GP}}$ | Tri | **0.00 ± 0.00** | **0.06 ± 0.02** | **−1.30 ± 0.18** |
| DHGP | Tri | 0.02 ± 0.00 | 0.14 ± 0.00 | −0.38 ± 0.10 |
| BNN (LC) | Tri | 15.06 ± 3.09 | 3.85 ± 0.40 | 472.28 ± 20.15 |
| BNN (orig) | Tri | 22.44 ± 3.04 | 4.72 ± 0.33 | 549.79 ± 44.15 |
| Ma$^{\text{GP}}$ | T1 | **0.01 ± 0.02** | **0.07 ± 0.08** | **−1.08 ± 0.95** |
| DHGP | T1 | 0.01 ± 0.00 | 0.09 ± 0.00 | −0.77 ± 0.02 |
| BNN (LC) | T1 | 15.10 ± 3.27 | 3.86 ± 0.39 | 442.54 ± 37.73 |
| BNN (orig) | T1 | 24.06 ± 4.84 | 4.88 ± 0.49 | 564.10 ± 90.25 |
| Ma$^{\text{GP}}$ | T1 small | 0.02 ± 0.00 | 0.12 ± 0.01 | **−0.54 ± 0.03** |
| DHGP | T1 small | 0.02 ± 0.00 | **0.12 ± 0.00** | −0.54 ± 0.06 |

# I  AbC Values as a Function of Computational Costs

Figure I.1 presents AbC values as a function of computational cost. From a cost perspective, *Smallest First* remains competitive with *Active Learning* when higher uncertainty is acceptable. Beyond $2 \cdot 10^5$ PetaFLOPs, its mean AbC value drops below that of *Active Learning*. In contrast, *Largest First* yields mean AbC values exceeding those of *Active Learning* after the third query, while *Random Order* exhibits high instability. Considering queries relative to compute costs, *Smallest First* enables more queries for the same PetaFLOP budget at the expense of higher uncertainty, whereas *Active Learning* achieves greater certainty at a higher cost.

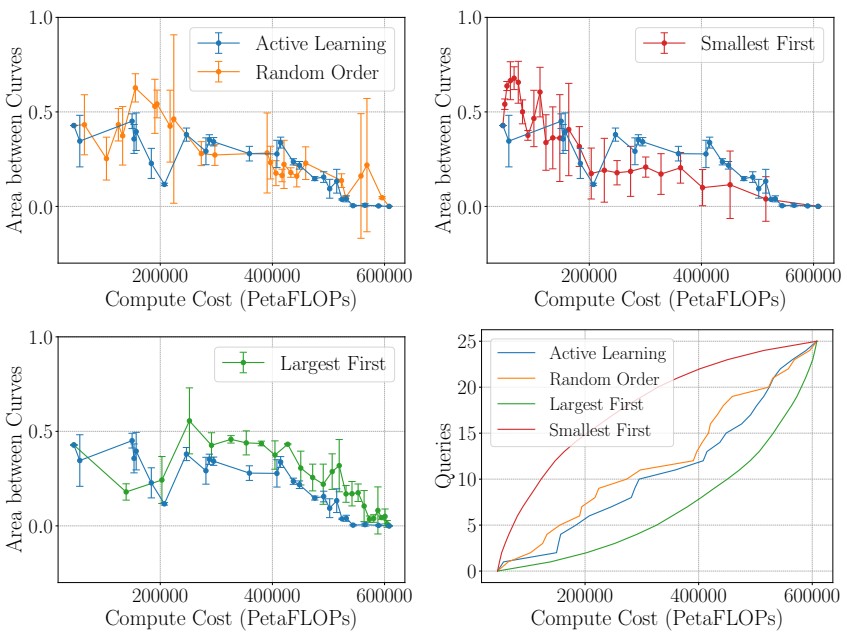

Figure I.1: AbC over cost in PetaFLOPs using various query strategies and a cost analysis.

# J  More Information about Query Order of used Query Strategies

Figure J.1 shows the order various query strategies were using when querying the next learning curve. 0 are the learning curves in the initial training dataset. Using the active learning strategy results in more certain learning curve predictions and translates into more certain scaling law predictions (compare main paper).

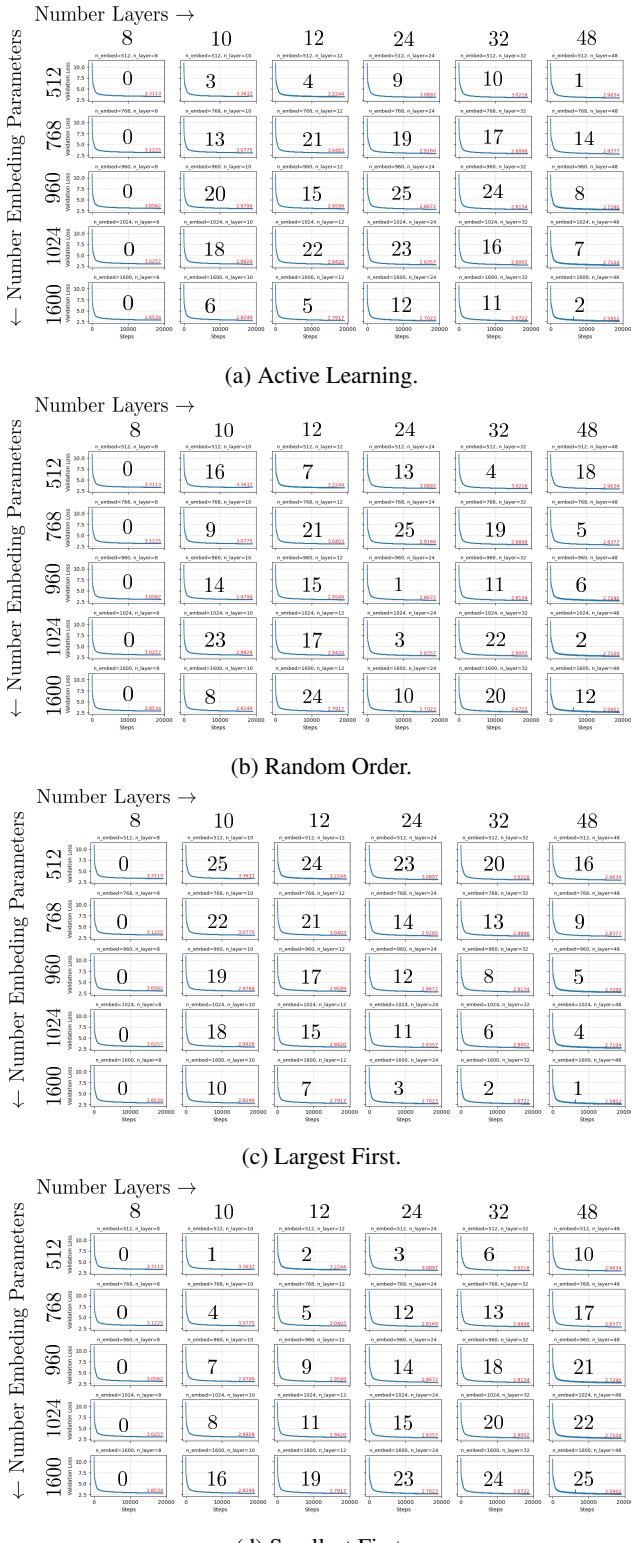

Figure J.1: Query order.

# K   Zero-Shot Performance Prediction for Bilingual Translation (Random Train-Test Split)

The train-test split in the main paper follows a structured pattern to ensure that each row and column of the dataset is missing exactly one language pair. In addition, we conducted experiments using a **random train-test** split. The zero-shot prediction performance under this setting is reported in Table K.1. A further analysis of the prediction performance for the last value and the last three values is provided in Table K.3 and Table K.2, respectively. All experimental settings were otherwise kept as in the main paper.
All results show that Ma$^{\mathrm{GP}}$ outperforms the baseline method DHGP.

Table K.1: Performance for zero-shot prediction on the bilingual dataset assuming a **random** train-test split. All metrics are computed on **all samples** of the entirely predicted learning curves.

| Method | Metric | Model | ↓RMSE | ↓MSE | ↓MAE | ↓MNLPD |
|---|---|---|---|---|---|---|
| Ma$^{\mathrm{GP}}$ | ChrF | mBart | **8.29 ± 1.71** | **83.94 ± 35.16** | **6.51 ± 1.48** | **6.26 ± 3.31** |
| DHGP | ChrF | mBart | 12.20 ± 2.16 | 179.34 ± 58.02 | 10.93 ± 2.05 | 8.61 ± 3.05 |
| Ma$^{\mathrm{GP}}$ | ChrF | Transformer | **4.79 ± 1.54** | **30.24 ± 23.21** | **3.72 ± 1.03** | **3.52 ± 0.72** |
| DHGP | ChrF | Transformer | 5.97 ± 1.30 | 44.07 ± 19.67 | 4.46 ± 0.81 | 4.75 ± 1.45 |
| Ma$^{\mathrm{GP}}$ | BLEU | mBart | **3.90 ± 1.48** | **22.16 ± 16.58** | **3.52 ± 1.49** | **7.01 ± 4.14** |
| DHGP | BLEU | mBart | 6.43 ± 1.17 | 61.07 ± 22.01 | 5.62 ± 1.00 | 18.03 ± 6.61 |
| Ma$^{\mathrm{GP}}$ | BLEU | Transformer | **1.45 ± 0.63** | **4.92 ± 4.16** | **0.95 ± 0.39** | **4.55 ± 5.68** |
| DHGP | BLEU | Transformer | 1.80 ± 0.30 | 5.21 ± 1.59 | 1.12 ± 0.15 | 22.29 ± 28.13 |

Table K.2: Performance for zero-shot prediction on the bilingual dataset assuming a **random** train-test split. All metrics are computed for the **last three** predicted values.

| Method | Metric | Model | ↓RMSE | ↓MSE | ↓MAE | ↓MNLPD |
|---|---|---|---|---|---|---|
| Ma$^{\mathrm{GP}}$ | ChrF | mBart | **4.88 ± 2.27** | **51.01 ± 48.57** | **4.40 ± 1.89** | **4.67 ± 1.88** |
| DHGP | ChrF | mBart | 9.78 ± 2.88 | 149.89 ± 88.37 | 9.42 ± 2.46 | 6.03 ± 2.84 |
| Ma$^{\mathrm{GP}}$ | ChrF | Transformer | **6.23 ± 2.98** | **65.01 ± 72.79** | **6.00 ± 2.93** | **5.12 ± 2.48** |
| DHGP | ChrF | Transformer | 8.88 ± 2.80 | 113.57 ± 69.29 | 8.52 ± 2.76 | 8.42 ± 5.03 |
| Ma$^{\mathrm{GP}}$ | BLEU | mBart | **3.65 ± 1.65** | **23.43 ± 17.11** | **3.60 ± 1.67** | **5.79 ± 3.41** |
| DHGP | BLEU | mBart | 7.90 ± 2.14 | 100.70 ± 42.13 | 7.86 ± 2.15 | 21.09 ± 10.79 |
| Ma$^{\mathrm{GP}}$ | BLEU | Transformer | **2.48 ± 1.17** | **14.22 ± 11.16** | **2.40 ± 1.16** | **11.98 ± 14.99** |
| DHGP | BLEU | Transformer | 2.99 ± 0.71 | 14.47 ± 5.59 | 2.74 ± 0.65 | 53.24 ± 83.49 |

Table K.3: Performance for zero-shot prediction on the bilingual dataset assuming a **random** train-test split. All metrics are computed for the **last** predicted value.

| Method | Metric | Model | ↓RMSE | ↓MSE | ↓MAE | ↓MNLPD |
|---|---|---|---|---|---|---|
| Ma$^{\mathrm{GP}}$ | ChrF | mBart | **4.01 ± 1.50** | **23.65 ± 17.76** | **4.01 ± 1.50** | **4.14 ± 1.69** |
| DHGP | ChrF | mBart | 8.84 ± 2.06 | 116.56 ± 52.88 | 8.84 ± 2.06 | 4.71 ± 1.16 |
| Ma$^{\mathrm{GP}}$ | ChrF | Transformer | **6.60 ± 3.55** | **82.83 ± 98.82** | **6.60 ± 3.55** | **6.00 ± 3.64** |
| DHGP | ChrF | Transformer | 9.37 ± 3.65 | 144.29 ± 103.84 | 9.37 ± 3.65 | 9.51 ± 5.95 |
| Ma$^{\mathrm{GP}}$ | BLEU | mBart | **3.67 ± 1.75** | **24.32 ± 18.25** | **3.67 ± 1.75** | **5.81 ± 3.42** |
| DHGP | BLEU | mBart | 8.05 ± 2.38 | 107.86 ± 48.53 | 8.05 ± 2.38 | 22.03 ± 13.63 |
| Ma$^{\mathrm{GP}}$ | BLEU | Transformer | **2.55 ± 1.12** | **14.06 ± 10.21** | **2.55 ± 1.12** | **12.23 ± 14.38** |
| DHGP | BLEU | Transformer | 3.12 ± 1.10 | 15.40 ± 9.46 | 3.12 ± 1.10 | 43.26 ± 69.56 |

## L Zero-Shot Performance Prediction for Bilingual Translation (Train-Test Split as in the Main Paper)

We provide a further analysis of the zero-shot performance prediction results for the bilingual dataset in this appendix.

Table L.1 presents the performance metrics for the full zero-shot predicted learning curve. The predicted performance on the last three data points and the last data point is reported in Table L.2 and Table L.3, respectively. Table L.1 demonstrates that Ma$^{GP}$ outperforms the baselines in terms of RMSE and MSE. DHGP achieves comparable performance in MAE when using learning curves obtained from the Transformer model with the ChrF metric. Although NBL exhibits the lowest uncertainty, its predictions yield the largest errors across all metrics. The error metrics on the last three and the last data point show that Ma$^{GP}$ outperforms all baselines in all error metrics, but showing increased MNLPD values compared to the NBL baseline. A visualization of the RMSE and MNLPD values for the zero-shot prediction of the last three data points and the last data point is additionally provided in Figure L.1 and Figure L.2, respectively. This demonstrates that, on average, when averaged over the learning curves in the test set, modeling a hierarchical structure and accounting for correlations among tasks is beneficial for zero-shot performance prediction in bilingual translation.

Table L.1: Performance for zero-shot prediction on the bilingual dataset for the train-test split used in the **main paper**. All metrics are computed on **all samples** of the entirely predicted learning curves.

| Method | Metric | Model | ↓RMSE | ↓MSE | ↓MAE | ↓MNLPD |
|---|---|---|---|---|---|---|
| Ma$^{GP}$ | ChrF | mBart | **7.02 ± 1.21** | **56.21 ± 21.24** | **5.84 ± 1.05** | 3.96 ± 0.55 |
| DHGP | ChrF | mBart | 9.54 ± 0.08 | 143.74 ± 3.06 | 8.75 ± 0.06 | 7.01 ± 0.76 |
| NBL | ChrF | mBart | 10.38 ± 3.99 | 123.76 ± 89.5 | 8.72 ± 3.43 | **3.77 ± 0.23** |
| Ma$^{GP}$ | ChrF | Transformer | **4.24 ± 0.63** | **19.98 ± 6.32** | 3.42 ± 0.54 | 3.27 ± 0.52 |
| DHGP | ChrF | Transformer | 4.38 ± 0.00 | 22.35 ± 0.01 | **3.37 ± 0.00** | 3.24 ± 0.00 |
| NBL | ChrF | Transformer | 6.16 ± 3.49 | 50.14 ± 48.39 | 4.30 ± 2.05 | **2.75 ± 0.14** |
| Ma$^{GP}$ | BLEU | mBart | **4.47 ± 1.09** | **33.61 ± 16.80** | **4.06 ± 1.07** | 5.93 ± 6.53 |
| DHGP | BLEU | mBart | 5.53 ± 0.00 | 49.13 ± 0.18 | 4.77 ± 0.01 | 13.28 ± 1.59 |
| NBL | BLEU | mBart | 5.94 ± 4.04 | 51.57 ± 56.89 | 5.13 ± 3.39 | **3.13 ± 0.45** |
| Ma$^{GP}$ | BLEU | Transformer | **1.29 ± 0.30** | **3.12 ± 2.47** | **0.89 ± 0.25** | 2.96 ± 1.74 |
| DHGP | BLEU | Transformer | 1.63 ± 0.00 | 3.51 ± 0.01 | 1.09 ± 0.00 | 5.84 ± 0.07 |
| NBL | BLEU | Transformer | 2.26 ± 1.23 | 6.63 ± 5.54 | 1.16 ± 0.67 | **−2.75 ± 0.65** |

Table L.2: Performance for zero-shot prediction on the bilingual dataset for the train-test split used in the **main paper**. All metrics are computed for the **last three** predicted values.

| Method | Metric | Model | ↓RMSE | ↓MSE | ↓MAE | ↓MNLPD |
|---|---|---|---|---|---|---|
| Ma$^{GP}$ | ChrF | mBart | **3.73 ± 0.16** | **20.50 ± 0.70** | **3.66 ± 0.14** | **3.29 ± 0.18** |
| DHGP | ChrF | mBart | 7.85 ± 0.09 | 97.79 ± 1.85 | 7.77 ± 0.09 | 5.19 ± 0.56 |
| NBL | ChrF | mBart | 11.1 ± 10.78 | 239.5 ± 319.75 | 10.63 ± 10.73 | 3.98 ± 0.51 |
| Ma$^{GP}$ | ChrF | Transformer | **5.58 ± 0.81** | **39.40 ± 11.64** | **5.39 ± 0.83** | 4.21 ± 0.93 |
| DHGP | ChrF | Transformer | 5.84 ± 0.01 | 53.14 ± 0.09 | 5.44 ± 0.01 | 4.21 ± 0.01 |
| NBL | ChrF | Transformer | 11.59 ± 8.17 | 201.05 ± 225.75 | 11.24 ± 8.34 | **3.96 ± 0.44** |
| Ma$^{GP}$ | BLEU | mBart | **3.99 ± 1.55** | **30.18 ± 19.14** | **3.95 ± 1.57** | 5.88 ± 7.21 |
| DHGP | BLEU | mBart | 7.75 ± 0.01 | 108.46 ± 0.110 | 7.70 ± 0.00 | 27.75 ± 3.73 |
| NBL | BLEU | mBart | 8.52 ± 7.47 | 128.33 ± 156.92 | 8.41 ± 7.39 | **3.69 ± 0.59** |
| Ma$^{GP}$ | BLEU | Transformer | **1.93 ± 0.60** | **9.36 ± 9.39** | **1.87 ± 0.61** | 7.19 ± 6.27 |
| DHGP | BLEU | Transformer | 2.82 ± 0.01 | 11.24 ± 0.06 | 2.63 ± 0.01 | 16.65 ± 0.22 |
| NBL | BLEU | Transformer | 4.79 ± 2.74 | 30.43 ± 26.8 | 4.61 ± 2.83 | **2.91 ± 0.56** |

Table L.3: Performance for zero-shot prediction on the bilingual dataset for the train-test split used in the **main paper**. All metrics are computed for the **last** predicted value.

| Method | Metric | Model | ↓RMSE | ↓MSE | ↓MAE | ↓MNLPD |
|---|---|---|---|---|---|---|
| Ma$^{GP}$ | ChrF | mBart | **3.64 ± 0.16** | **19.88 ± 2.10** | **3.64 ± 0.16** | **3.19 ± 0.19** |
| DHGP | ChrF | mBart | 7.53 ± 0.10 | 93.63 ± 1.72 | 7.53 ± 0.10 | 5.14 ± 0.53 |
| NBL | ChrF | mBart | 13.23 ± 13.85 | 367.05 ± 490.53 | 13.23 ± 13.85 | 4.10 ± 0.65 |
| Ma$^{GP}$ | ChrF | Transformer | **6.29 ± 0.84** | **51.33 ± 15.49** | **6.29 ± 0.84** | 4.72 ± 1.25 |
| DHGP | ChrF | Transformer | 6.82 ± 0.01 | 91.06 ± 0.09 | 6.82 ± 0.01 | 5.72 ± 0.02 |
| NBL | ChrF | Transformer | 13.19 ± 8.99 | 254.96 ± 283.48 | 13.19 ± 8.99 | **4.11 ± 0.42** |
| Ma$^{GP}$ | BLEU | mBart | **3.91 ± 1.56** | **30.37 ± 19.66** | **3.91 ± 1.56** | 5.63 ± 7.04 |
| DHGP | BLEU | mBart | 8.14 ± 0.01 | 120.50 ± 0.210 | 8.14 ± 0.01 | 33.29 ± 4.56 |
| NBL | BLEU | mBart | 9.56 ± 9.26 | 177.08 ± 226.57 | 9.56 ± 9.26 | **3.87 ± 0.77** |
| Ma$^{GP}$ | BLEU | Transformer | **1.87 ± 0.49** | **8.79 ± 8.20** | **1.87 ± 0.49** | 6.23 ± 5.10 |
| DHGP | BLEU | Transformer | 3.33 ± 0.01 | 17.66 ± 0.07 | 3.33 ± 0.01 | 20.59 ± 0.25 |
| NBL | BLEU | Transformer | 5.62 ± 3.23 | 42.00 ± 32.91 | 5.62 ± 3.23 | **3.24 ± 0.43** |

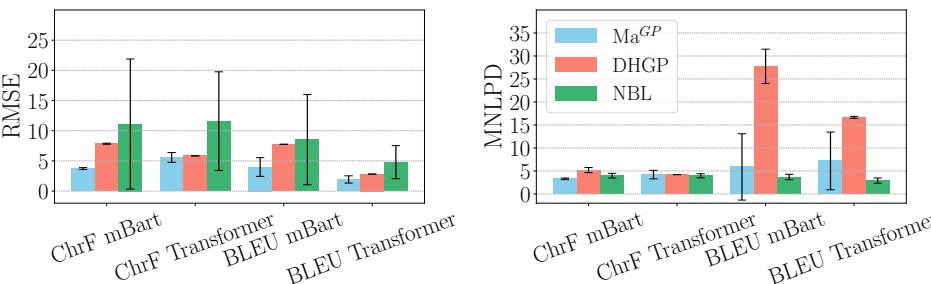

Figure L.1: Performance for zero-shot prediction on the bilingual dataset for the train-test split used in the **main paper**. All metrics are computed for the **last three** predicted values.

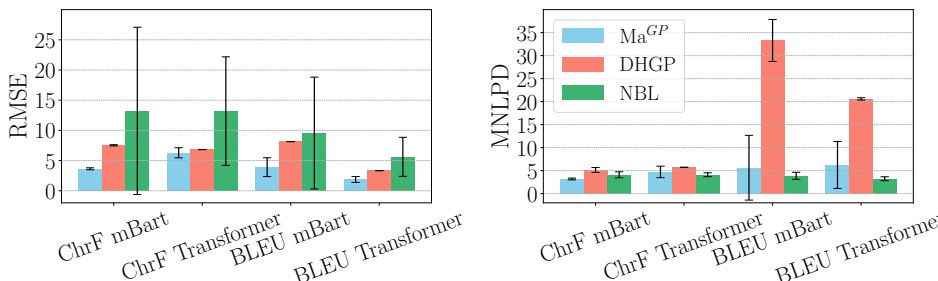

Figure L.2: Performance for zero-shot prediction on the bilingual dataset for the train-test split used in the **main paper**. All metrics are computed for the **last** predicted value.

Additionally, Figure L.3 and Figure L.4 show the individual performance predictions for each learning curve, which corresponds to the translation performance from a source into a target language, using BLEU and ChrF as evaluation metrics, respectively. Figure L.3 shows, that while Ma$^{GP}$ yields RMSE values higher than at least one baseline using the mBART50 (mB) dataset and the language pairs *id-jv*, *en-id*, and *jv-ms*, it performs comparably to the baselines using the Transformer (T) dataset, underperforming only for *en-id* and *jv-ms*. Similarly, Figure L.4 demonstrates, that Ma$^{GP}$ underperforms the baselines for the language pairs *id-jv*, *en-id*, and *jv-ms* using the mBART50 (mB) dataset, and for *jv-ms* and *ta-tl* with the Transformer (T) dataset.

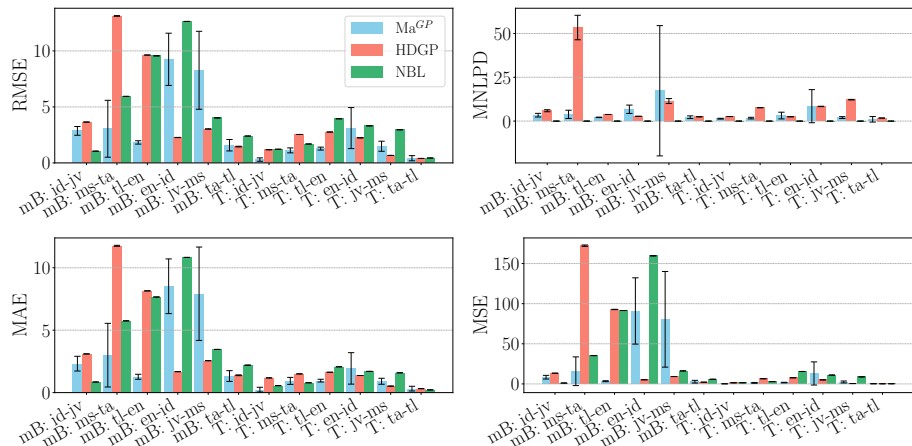

Figure L.3: Performance for zero-shot prediction on the bilingual dataset for the train-test split used in the **main paper**. All metrics are computed on **all samples** of the entirely predicted learning curves using either mBART50 (mB) or the Transformer (T) dataset. Performance prediction was done for the **metric BLEU**.

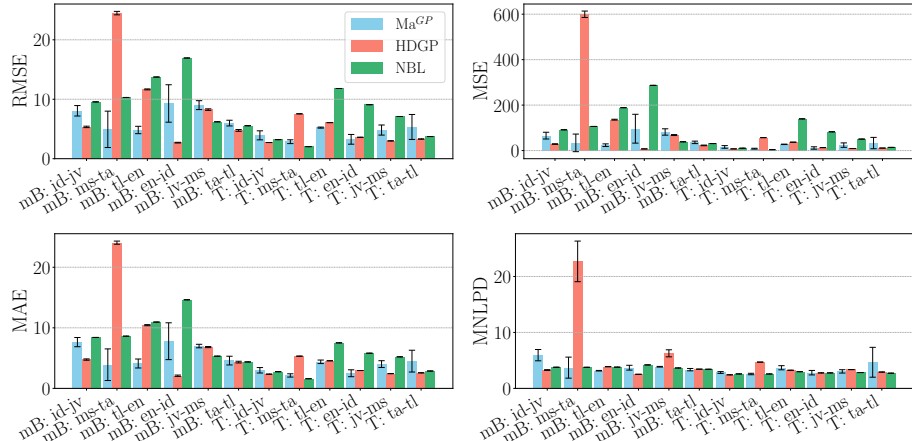

Figure L.4: Performance for zero-shot prediction on the bilingual dataset for the train-test split used in the **main paper**. All metrics are computed on **all samples** of the entirely predicted learning curves using either mBART50 (mB) or the Transformer (T) dataset. Performance prediction was done for the **metric ChrF**.

## M    Exchanged Hierarchies on the Bilingual Dataset

To validate the exchangeability of hierarchies, we present additional results for the Ma$^{GP}$ and DHGP models in Figure M.1, where target languages are assigned to the upper hierarchy level and source languages to the lower level. As shown, Ma$^{GP}$ significantly outperforms DHGP in terms of RMSE under this assumption. Furthermore, a comparison of both hierarchical assumptions in Figure M.2 indicates that the exchanged hierarchy yields a slight RMSE improvement. This suggests that translations into a common target language share more information suitable for the upper hierarchical level with significantly reduced uncertainty.

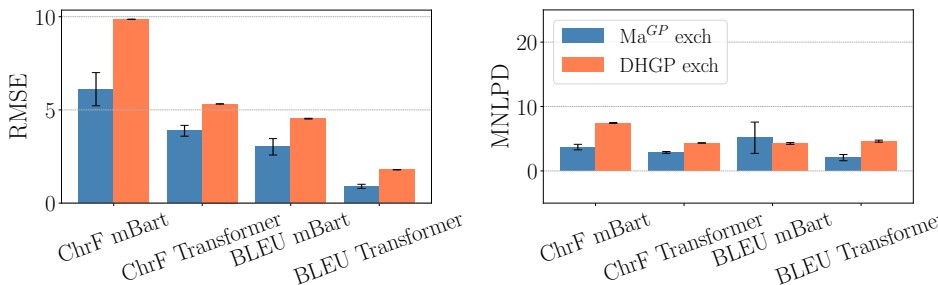

Figure M.1: Zero-shot learning curve prediction results on the bilingual dataset assuming the exchanged (exch) hierarchy: target languages over sources languages.

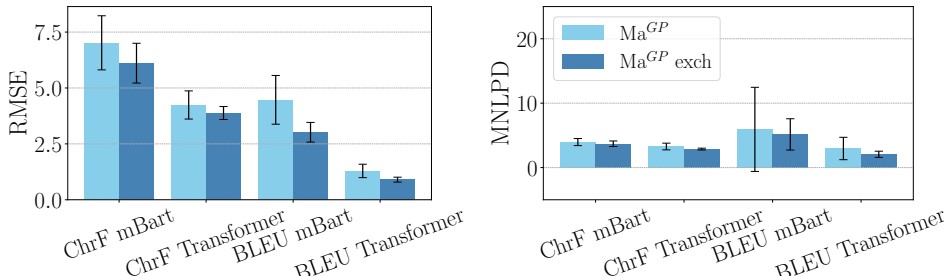

Figure M.2: Zero-shot learning curve prediction results on the bilingual dataset. Comparing the RMSE values of both hierarchical assumptions: source languages over target languages and the exchanged version (exch).

## N    Few-Shot Performance Prediction for Architecture Design

To offer another experimental setting to confirm our *core hypothesis* that learning curve datasets exhibit hierarchical structures, in this section we provide experiments for few-shot performance prediction of learning curves across various model architectures obtained from multilingual translation.

We assume that learning curves of already fine-tuned M2M100 models of sizes 175M and 615M are available. Based on these, we predict (i.e., extrapolate) the final nine missing data points of the learning curves for the 1.2B model, which correspond to performance values at the largest dataset sizes. Accordingly, we assess whether acquiring additional data for translation into a specific target language or from a particular source language is justified based on the predicted performance. To this end, we utilize the learning curves of the smaller models alongside the initial performance values of the largest model to extrapolate the final data points.

**Exchanged Hierarchies on the Multilingual Dataset**

Figure N.1 shows the extrapolation performance when predicting the performance for a translation from a common **source** language. Similar to predicting the performance into a common target language (main paper), using $\text{Ma}^{\text{GP}}$, we consistently outperform the baseline models in terms of RMSE, despite higher predictive uncertainty. The highest uncertainty is observed for learning curves predicting performance over dataset size for Tagalog (*tl*) and Tamil (*ta*) as a source languages. Nevertheless, mean predictions across all languages and translation scenarios achieve RMSE values superior to the baselines. These results confirm that incorporating hierarchical assumptions and leveraging correlations among tasks enhances prediction performance. Furthermore, this confirms our additional hypothesis, that hierarchies are exchangeable. The numerical values for these Figures are given in Table N.1.

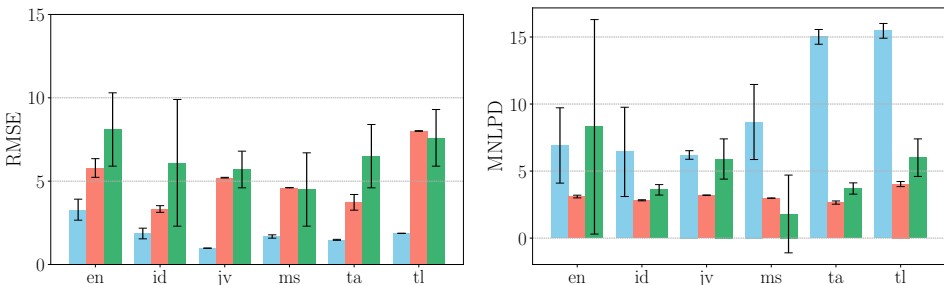

Figure N.1: Few-shot performance prediction on the multilingual dataset, extrapolating the last nine data points of each learning curve for the M2M100 1.2B model. The extrapolated learning curves show performance over dataset size for translations from a common **source** language.

## Extended Analysis

The RMSE extrapolation performance for the last 3 and 1 values are demonstrated in Figure N.2 and Figure N.3. Additionally, all performance metrics are provided in Table N.2 and Table N.3, respectively. In these scenarios, $Ma^{GP}$ does not consistently outperform the baselines for every individual source or target language, but its performance remains broadly comparable. These experiments underline the particular strength of the proposed framework in situations where multiple data points require extrapolation, as demonstrated in Figure N.1 and Figure 9 in the main paper. In those settings, $Ma^{GP}$ substantially outperforms all baseline methods.

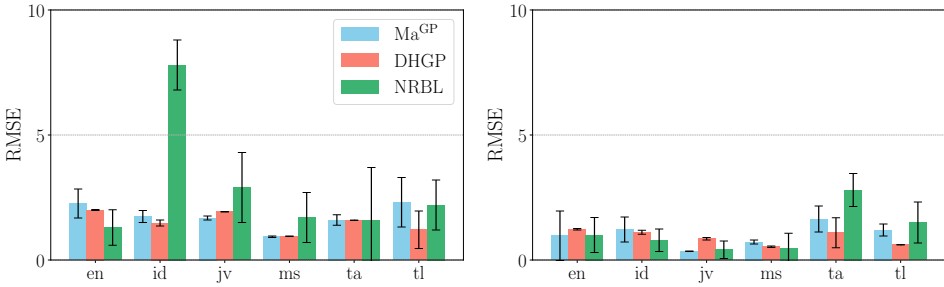

Figure N.2: Few-shot performance prediction on the multilingual dataset, extrapolating learning curves for the M2M100 1.2B model. The extrapolated learning curves show performance over dataset size for translations into a common **target** language.
Left: Extrapolation of the last three data points. Right: Extrapolation of the final data point.

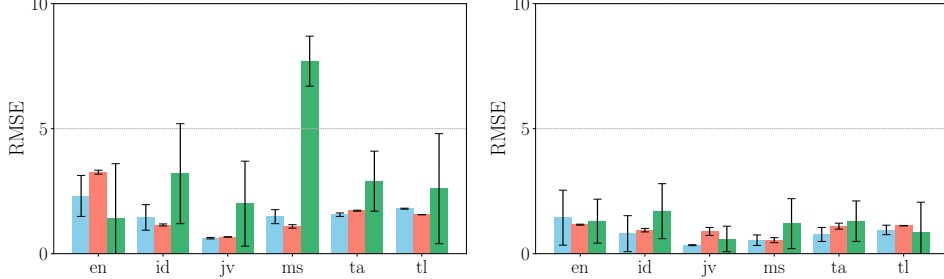

Figure N.3: Few-shot performance prediction on the multilingual dataset, extrapolating learning curves for the M2M100 1.2B model. The extrapolated learning curves show performance over dataset size for translations from a common **source** language shown on the x-axis.
Left: Extrapolation of the last three data points. Right: Extrapolation of the last data point.

Table N.1: Few-shot performance prediction on the multilingual dataset assuming we are interested in the extrapolation of the last **9** missing values of learning curves for the 1.2B model. The results show the extrapolation performances for learning curves of the multilingual translation from a common source or into a common target language.

| Method | Language | ↓RMSE | ↓MSE | ↓MAE | ↓MNLPD |
|---|---|---|---|---|---|
| Ma$^{GP}$ | Source: en | **3.29 ± 0.63** | 12.17 ± 4.41 | **2.89 ± 0.61** | 6.91 ± 2.81 |
| DHGP | Source: en | 5.79 ± 0.56 | 45.72 ± 7.59 | 4.81 ± 0.47 | **3.10 ± 0.10** |
| NRBL | Source: en | 8.10 ± 2.20 | **7.70 ± 2.20** | 7.00 ± 3.50 | 8.30 ± 8.00 |
| Ma$^{GP}$ | Source: id | **1.86 ± 0.32** | **4.00 ± 1.10** | **1.60 ± 0.32** | 6.43 ± 3.33 |
| DHGP | Source: id | 3.33 ± 0.20 | 15.92 ± 1.60 | 3.02 ± 0.20 | **2.82 ± 0.04** |
| NRBL | Source: id | 6.10 ± 3.80 | 5.80 ± 3.90 | 5.30 ± 5.80 | 3.60 ± 0.39 |
| Ma$^{GP}$ | Source: jv | **0.98 ± 0.01** | **1.09 ± 0.01** | **0.74 ± 0.01** | 6.20 ± 0.32 |
| DHGP | Source: jv | 5.21 ± 0.02 | 31.01 ± 0.15 | 4.38 ± 0.01 | **3.20 ± 0.02** |
| NRBL | Source: jv | 5.70 ± 1.10 | 4.90 ± 1.00 | 3.40 ± 1.20 | 5.90 ± 1.50 |
| Ma$^{GP}$ | Source: ms | **1.68 ± 0.10** | **3.01 ± 0.33** | **1.46 ± 0.10** | 8.66 ± 2.80 |
| DHGP | Source: ms | 4.61 ± 0.00 | 30.10 ± 0.02 | 4.15 ± 0.00 | 2.98 ± 0.00 |
| NRBL | Source: ms | 4.50 ± 2.20 | 4.10 ± 2.00 | 2.50 ± 2.00 | **1.80 ± 2.90** |
| Ma$^{GP}$ | Source: ta | **1.48 ± 0.03** | **2.31 ± 0.07** | **1.16 ± 0.02** | 15.01 ± 0.55 |
| DHGP | Source: ta | 3.73 ± 0.47 | 17.32 ± 3.64 | 3.20 ± 0.40 | **2.65 ± 0.12** |
| NRBL | Source: ta | 6.50 ± 1.90 | 5.80 ± 1.70 | 4.60 ± 2.20 | 3.70 ± 0.42 |
| Ma$^{GP}$ | Source: tl | **1.87 ± 0.01** | **3.52 ± 0.04** | **1.58 ± 0.01** | 15.46 ± 0.55 |
| DHGP | Source: tl | 8.01 ± 0.02 | 72.42 ± 0.48 | 7.26 ± 0.02 | **4.03 ± 0.19** |
| NRBL | Source: tl | 7.60 ± 1.70 | 7.00 ± 1.70 | 6.10 ± 2.40 | 6.00 ± 1.40 |
| Ma$^{GP}$ | Target: en | **2.11 ± 0.19** | **4.81 ± 0.74** | **1.82 ± 0.19** | 11.09 ± 7.94 |
| DHGP | Target: en | 2.90 ± 0.15 | 12.24 ± 0.40 | 2.52 ± 0.19 | **2.57 ± 0.02** |
| NRBL | Target: en | 7.20 ± 4.10 | 6.70 ± 4.10 | 6.80 ± 5.80 | 5.20 ± 1.50 |
| Ma$^{GP}$ | Target: id | **1.66 ± 0.15** | **3.21 ± 0.79** | **1.43 ± 0.15** | 20.50 ± 17.62 |
| DHGP | Target: id | 3.42 ± 0.12 | 13.03 ± 0.83 | 2.83 ± 0.10 | 2.61 ± 0.06 |
| NRBL | Target: id | 7.30 ± 2.50 | 6.90 ± 2.60 | 6.00 ± 3.80 | **2.00 ± 2.90** |
| Ma$^{GP}$ | Target: jv | **1.43 ± 0.11** | **2.06 ± 0.32** | **1.19 ± 0.05** | 9.96 ± 3.77 |
| DHGP | Target: jv | 2.12 ± 0.41 | 5.42 ± 2.32 | 1.81 ± 0.39 | **2.16 ± 0.29** |
| NRBL | Target: jv | 3.60 ± 1.10 | 3.40 ± 1.10 | 1.40 ± 8.10 | 8.20 ± 8.10 |
| Ma$^{GP}$ | Target: ms | **0.92 ± 0.01** | **0.90 ± 0.01** | **0.78 ± 0.01** | 6.50 ± 0.29 |
| DHGP | Target: ms | 3.00 ± 0.64 | 11.02 ± 3.31 | 2.47 ± 0.48 | **2.30 ± 0.12** |
| NRBL | Target: ms | 5.60 ± 1.60 | 4.90 ± 1.40 | 3.40 ± 1.60 | 4.80 ± 1.30 |
| Ma$^{GP}$ | Target: ta | **2.50 ± 0.01** | **6.51 ± 0.03** | **1.78 ± 0.00** | 19.21 ± 0.37 |
| DHGP | Target: ta | 11.21 ± 0.17 | 128.25 ± 3.99 | 10.02 ± 0.15 | 4.62 ± 0.12 |
| NRBL | Target: ta | 7.70 ± 1.20 | 7.10 ± 1.20 | 6.10 ± 2.00 | **3.80 ± 0.32** |
| Ma$^{GP}$ | Target: tl | **3.36 ± 1.09** | 13.10 ± 8.25 | **2.93 ± 0.93** | 67.88 ± 38.98 |
| DHGP | Target: tl | 7.84 ± 0.28 | 65.56 ± 4.32 | 7.16 ± 0.26 | **3.84 ± 0.18** |
| NRBL | Target: tl | 7.10 ± 0.60 | **6.30 ± 0.62** | 5.10 ± 8.70 | 4.00 ± 0.72 |

Table N.2: Few-shot performance prediction on the multilingual dataset assuming we are interested in the extrapolation of the last **3** missing values of learning curves for the 1.2B model. The results show the extrapolation performances for learning curves of the multilingual translation from a common source (Source) or into a common target (Target) language.

| Method | Language | ↓RMSE | ↓MSE | ↓MAE | ↓MNLPD |
|---|---|---|---|---|---|
| Ma$^{GP}$ | Source: en | $2.31 \pm 0.82$ | $7.30 \pm 3.96$ | $2.16 \pm 0.87$ | $\mathbf{2.73 \pm 0.15}$ |
| DHGP | Source: en | $3.26 \pm 0.08$ | $19.67 \pm 1.97$ | $3.02 \pm 0.09$ | $3.15 \pm 0.12$ |
| NRBL | Source: en | $\mathbf{1.40 \pm 2.20}$ | $\mathbf{1.40 \pm 2.30}$ | $\mathbf{0.07 \pm 0.00}$ | $7.00 \pm 0.01$ |
| Ma$^{GP}$ | Source: id | $1.45 \pm 0.51$ | $2.81 \pm 1.81$ | $1.38 \pm 0.52$ | $1.99 \pm 0.24$ |
| DHGP | Source: id | $\mathbf{1.15 \pm 0.04}$ | $\mathbf{2.43 \pm 0.06}$ | $\mathbf{1.02 \pm 0.05}$ | $\mathbf{1.79 \pm 0.01}$ |
| NRBL | Source: id | $3.20 \pm 2.00$ | $3.20 \pm 2.00$ | $1.40 \pm 1.90$ | $2.70 \pm 0.49$ |
| Ma$^{GP}$ | Source: jv | $\mathbf{0.62 \pm 0.03}$ | $\mathbf{0.50 \pm 0.03}$ | $\mathbf{0.53 \pm 0.02}$ | $\mathbf{1.12 \pm 0.10}$ |
| DHGP | Source: jv | $0.67 \pm 0.01$ | $0.59 \pm 0.02$ | $0.62 \pm 0.00$ | $1.31 \pm 0.01$ |
| NRBL | Source: jv | $2.00 \pm 1.70$ | $1.90 \pm 1.70$ | $6.70 \pm 8.20$ | $2.90 \pm 0.47$ |
| Ma$^{GP}$ | Source: ms | $1.48 \pm 0.28$ | $2.56 \pm 0.80$ | $1.41 \pm 0.26$ | $3.26 \pm 1.08$ |
| DHGP | Source: ms | $\mathbf{1.09 \pm 0.07}$ | $\mathbf{1.56 \pm 0.12}$ | $1.02 \pm 0.06$ | $\mathbf{1.76 \pm 0.11}$ |
| NRBL | Source: ms | $7.70 \pm 1.00$ | $7.70 \pm 1.00$ | $\mathbf{0.02 \pm 0.03}$ | $4.50 \pm 2.60$ |
| Ma$^{GP}$ | Source: ta | $\mathbf{1.56 \pm 0.07}$ | $\mathbf{2.85 \pm 0.32}$ | $1.42 \pm 0.07$ | $9.49 \pm 8.14$ |
| DHGP | Source: ta | $1.72 \pm 0.02$ | $3.81 \pm 0.05$ | $1.62 \pm 0.01$ | $\mathbf{2.51 \pm 0.01}$ |
| NRBL | Source: ta | $2.90 \pm 1.20$ | $2.80 \pm 1.30$ | $\mathbf{1.00 \pm 6.20}$ | $2.80 \pm 0.22$ |
| Ma$^{GP}$ | Source: tl | $1.80 \pm 0.02$ | $3.74 \pm 0.02$ | $1.76 \pm 0.00$ | $5.48 \pm 2.63$ |
| DHGP | Source: tl | $\mathbf{1.56 \pm 0.00}$ | $3.08 \pm 0.03$ | $1.42 \pm 0.01$ | $\mathbf{2.00 \pm 0.01}$ |
| NRBL | Source: tl | $2.60 \pm 2.20$ | $\mathbf{2.60 \pm 2.30}$ | $\mathbf{1.20 \pm 1.50}$ | $2.90 \pm 0.44$ |
| Ma$^{GP}$ | Target: en | $2.26 \pm 0.58$ | $5.74 \pm 2.29$ | $2.15 \pm 0.62$ | $11.67 \pm 11.42$ |
| DHGP | Target: en | $2.00 \pm 0.02$ | $4.95 \pm 0.06$ | $\mathbf{1.80 \pm 0.01}$ | $\mathbf{2.05 \pm 0.00}$ |
| NRBL | Target: en | $\mathbf{1.30 \pm 0.71}$ | $\mathbf{1.20 \pm 0.77}$ | $2.30 \pm 2.10$ | $3.80 \pm 2.70$ |
| Ma$^{GP}$ | Target: id | $1.74 \pm 0.24$ | $3.77 \pm 0.54$ | $1.65 \pm 0.24$ | $5.63 \pm 4.31$ |
| DHGP | Target: id | $\mathbf{1.48 \pm 0.12}$ | $\mathbf{3.42 \pm 0.18}$ | $1.39 \pm 0.11$ | $\mathbf{2.40 \pm 0.11}$ |
| NRBL | Target: id | $7.80 \pm 1.00$ | $7.70 \pm 1.00$ | $\mathbf{0.02 \pm 0.03}$ | $7.00 \pm 0.01$ |
| Ma$^{GP}$ | Target: jv | $\mathbf{1.68 \pm 0.08}$ | $3.04 \pm 0.26$ | $1.53 \pm 0.06$ | $4.60 \pm 1.25$ |
| DHGP | Target: jv | $1.93 \pm 0.01$ | $5.26 \pm 0.01$ | $1.84 \pm 0.01$ | $3.29 \pm 0.02$ |
| NRBL | Target: jv | $2.90 \pm 1.40$ | $\mathbf{2.80 \pm 1.40}$ | $\mathbf{1.00 \pm 8.40}$ | $2.50 \pm 0.49$ |
| Ma$^{GP}$ | Target: ms | $\mathbf{0.93 \pm 0.03}$ | $\mathbf{1.12 \pm 0.07}$ | $0.89 \pm 0.02$ | $3.73 \pm 1.03$ |
| DHGP | Target: ms | $0.95 \pm 0.01$ | $1.44 \pm 0.06$ | $\mathbf{0.89 \pm 0.01}$ | $1.73 \pm 0.02$ |
| NRBL | Target: ms | $1.70 \pm 1.00$ | $1.70 \pm 1.00$ | $4.00 \pm 4.70$ | $2.80 \pm 0.05$ |
| Ma$^{GP}$ | Target: ta | $1.60 \pm 0.21$ | $3.62 \pm 0.13$ | $1.41 \pm 0.26$ | $4.12 \pm 1.79$ |
| DHGP | Target: ta | $\mathbf{1.59 \pm 0.00}$ | $2.75 \pm 0.02$ | $1.36 \pm 0.02$ | $\mathbf{2.19 \pm 0.01}$ |
| NRBL | Target: ta | $1.60 \pm 2.10$ | $\mathbf{1.60 \pm 2.10}$ | $\mathbf{0.07 \pm 0.01}$ | $3.90 \pm 1.00$ |
| Ma$^{GP}$ | Target: tl | $2.31 \pm 0.99$ | $6.94 \pm 4.34$ | $2.18 \pm 0.88$ | $28.23 \pm 28.66$ |
| DHGP | Target: tl | $\mathbf{1.21 \pm 0.75}$ | $2.25 \pm 2.24$ | $\mathbf{1.05 \pm 0.67}$ | $\mathbf{1.62 \pm 0.37}$ |
| NRBL | Target: tl | $2.20 \pm 1.00$ | $\mathbf{2.20 \pm 1.00}$ | $6.00 \pm 3.80$ | $2.80 \pm 0.13$ |

Table N.3: Few-shot performance prediction on the multilingual dataset assuming we are interested in the extrapolation of the last **1** missing values of learning curves for the 1.2B model. The results show the extrapolation performances for learning curves of the multilingual translation from a common source (Source) or into a common target (Target) language.

| Method | Language | ↓RMSE | ↓MSE | ↓MAE | ↓MNLPD |
|---|---|---|---|---|---|
| Ma$^{GP}$ | Source: en | $1.44 \pm 1.10$ | $3.88 \pm 4.44$ | $1.44 \pm 1.10$ | $1.85 \pm 0.54$ |
| DHGP | Source: en | $\mathbf{1.16 \pm 0.02}$ | $2.06 \pm 0.05$ | $\mathbf{1.16 \pm 0.02}$ | $\mathbf{1.79 \pm 0.01}$ |
| NRBL | Source: en | $1.30 \pm 0.88$ | $\mathbf{1.30 \pm 0.88}$ | $2.40 \pm 2.70$ | $2.90 \pm 1.60$ |
| Ma$^{GP}$ | Source: id | $\mathbf{0.80 \pm 0.72}$ | $1.28 \pm 2.01$ | $\mathbf{0.80 \pm 0.72}$ | $\mathbf{1.29 \pm 0.51}$ |
| DHGP | Source: id | $0.94 \pm 0.07$ | $\mathbf{1.13 \pm 0.13}$ | $0.94 \pm 0.07$ | $1.49 \pm 0.05$ |
| NRBL | Source: id | $1.70 \pm 1.10$ | $1.70 \pm 1.10$ | $4.10 \pm 3.90$ | $4.90 \pm 5.20$ |
| Ma$^{GP}$ | Source: jv | $\mathbf{0.34 \pm 0.02}$ | $\mathbf{0.14 \pm 0.01}$ | $\mathbf{0.34 \pm 0.02}$ | $\mathbf{0.54 \pm 0.03}$ |
| DHGP | Source: jv | $0.89 \pm 0.16$ | $1.09 \pm 0.28$ | $0.89 \pm 0.16$ | $1.51 \pm 0.18$ |
| NRBL | Source: jv | $0.59 \pm 0.51$ | $0.59 \pm 0.51$ | $0.61 \pm 0.83$ | $1.70 \pm 0.33$ |
| Ma$^{GP}$ | Source: ms | $0.54 \pm 0.21$ | $\mathbf{0.40 \pm 0.26}$ | $0.54 \pm 0.21$ | $\mathbf{1.22 \pm 0.70}$ |
| DHGP | Source: ms | $\mathbf{0.54 \pm 0.10}$ | $0.45 \pm 0.23$ | $\mathbf{0.54 \pm 0.10}$ | $1.25 \pm 0.19$ |
| NRBL | Source: ms | $1.20 \pm 1.00$ | $1.20 \pm 1.00$ | $2.60 \pm 3.70$ | $1.70 \pm 0.70$ |
| Ma$^{GP}$ | Source: ta | $\mathbf{0.77 \pm 0.28}$ | $\mathbf{0.75 \pm 0.51}$ | $\mathbf{0.77 \pm 0.28}$ | $2.46 \pm 1.74$ |
| DHGP | Source: ta | $1.10 \pm 0.12$ | $1.62 \pm 0.32$ | $1.10 \pm 0.12$ | $\mathbf{1.70 \pm 0.13}$ |
| NRBL | Source: ta | $1.30 \pm 0.81$ | $1.30 \pm 0.81$ | $2.40 \pm 2.20$ | $1.90 \pm 0.49$ |
| Ma$^{GP}$ | Source: tl | $0.95 \pm 0.19$ | $\mathbf{1.09 \pm 0.34}$ | $0.95 \pm 0.19$ | $\mathbf{1.66 \pm 0.36}$ |
| DHGP | Source: tl | $1.12 \pm 0.01$ | $1.66 \pm 0.00$ | $1.12 \pm 0.01$ | $1.67 \pm 0.00$ |
| NRBL | Source: tl | $\mathbf{0.86 \pm 1.20}$ | $0.86 \pm 1.20$ | $2.10 \pm 3.90$ | $1.80 \pm 0.53$ |
| Ma$^{GP}$ | Target: en | $\mathbf{0.97 \pm 0.99}$ | $2.03 \pm 3.54$ | $\mathbf{0.97 \pm 0.99}$ | $\mathbf{1.67 \pm 0.85}$ |
| DHGP | Target: en | $1.23 \pm 0.03$ | $1.73 \pm 0.08$ | $1.23 \pm 0.03$ | $1.74 \pm 0.04$ |
| NRBL | Target: en | $1.00 \pm 0.70$ | $\mathbf{1.00 \pm 0.70}$ | $1.60 \pm 1.80$ | $4.40 \pm 5.50$ |
| Ma$^{GP}$ | Target: id | $1.22 \pm 0.50$ | $1.93 \pm 1.09$ | $1.22 \pm 0.50$ | $\mathbf{1.66 \pm 0.36}$ |
| DHGP | Target: id | $1.11 \pm 0.08$ | $2.09 \pm 0.22$ | $1.11 \pm 0.08$ | $2.16 \pm 0.17$ |
| NRBL | Target: id | $\mathbf{0.79 \pm 0.45}$ | $\mathbf{0.70 \pm 0.45}$ | $\mathbf{0.83 \pm 0.78}$ | $2.40 \pm 1.80$ |
| Ma$^{GP}$ | Target: jv | $\mathbf{0.35 \pm 0.01}$ | $\mathbf{0.21 \pm 0.01}$ | $0.35 \pm 0.01$ | $\mathbf{0.75 \pm 0.05}$ |
| DHGP | Target: jv | $0.85 \pm 0.05$ | $0.89 \pm 0.09$ | $0.85 \pm 0.05$ | $1.36 \pm 0.05$ |
| NRBL | Target: jv | $0.41 \pm 0.35$ | $0.41 \pm 0.35$ | $\mathbf{0.29 \pm 0.35}$ | $1.50 \pm 0.76$ |
| Ma$^{GP}$ | Target: ms | $0.72 \pm 0.08$ | $0.60 \pm 0.08$ | $0.72 \pm 0.08$ | $1.40 \pm 0.12$ |
| DHGP | Target: ms | $0.53 \pm 0.03$ | $\mathbf{0.45 \pm 0.04}$ | $\mathbf{0.53 \pm 0.03}$ | $\mathbf{1.03 \pm 0.04}$ |
| NRBL | Target: ms | $\mathbf{0.48 \pm 0.59}$ | $0.48 \pm 0.59$ | $0.59 \pm 1.00$ | $1.70 \pm 0.25$ |
| Ma$^{GP}$ | Target: ta | $1.64 \pm 0.52$ | $3.32 \pm 1.67$ | $1.64 \pm 0.52$ | $8.72 \pm 5.24$ |
| DHGP | Target: ta | $\mathbf{1.09 \pm 0.60}$ | $\mathbf{1.72 \pm 1.33}$ | $\mathbf{1.09 \pm 0.60}$ | $\mathbf{1.67 \pm 0.31}$ |
| NRBL | Target: ta | $2.80 \pm 0.66$ | $2.80 \pm 0.66$ | $8.10 \pm 3.10$ | $2.70 \pm 0.26$ |
| Ma$^{GP}$ | Target: tl | $1.20 \pm 0.24$ | $2.17 \pm 0.75$ | $1.20 \pm 0.24$ | $3.89 \pm 2.63$ |
| DHGP | Target: tl | $\mathbf{0.61 \pm 0.00}$ | $\mathbf{0.44 \pm 0.00}$ | $\mathbf{0.61 \pm 0.00}$ | $\mathbf{1.39 \pm 0.00}$ |
| NRBL | Target: tl | $1.50 \pm 0.82$ | $1.50 \pm 0.82$ | $2.90 \pm 2.30$ | $2.20 \pm 0.21$ |

## O    Naive Regression Baseline for Few-Shot Prediction

We define a naive regression baseline (NRBL) as a non-linear regression model trained on the available initial training data points of the 1.2B model, together with the complete learning curves of the 175M and 615M models. This model fits several functional forms to the training data, including a vapor-pressure function, the MMF function, a power law function, and a logarithmic function. These functions are given by

$$f_{\text{vapor}}(x) = \exp\left(a + \frac{b}{x} + c\log x\right),$$
$$f_{\text{MMF}}(x) = \frac{ab + cx}{b + x},$$
$$f_{\text{power}}(x) = a\,x^b + c,$$
$$f_{\text{log}}(x) = c + a\log(bx).$$

The reported performance is the average across these fitted models.

## P    On Compute Resources and Complexity

The Ma$^{\text{GP}}$ framework and DHGP model were trained and evaluated on an Intel® Core™ i7-8565U CPU. Baseline methods were trained and evaluated on an Intel® Xeon® Gold 6448H CPU.

Ma$^{\text{GP}}$ is implemented using the inducing points method. Ma$^{\text{GP}}$ is originally referred to as HMOGP-LV Ma et al. [24]. HMOGP-LV is derived from LVMOGP Dai et al. [111]. The computational complexity is

$$\mathcal{O}\left(\max(NR,\ M_H)\cdot\max(D,\ M_X)\cdot\max(M_H,\ M_X)\right),$$

with $N$ being the available data points per learning curve, $R$ being the number of learning curves per task ($d$), $D$ being the number of tasks $t$, $M_X = M_r \times R$ for $M_r$ being the number of inducing input points in the $r^{\text{th}}$ learning curve of a task and $M_H$ being the number of inducing output points.

The wall-clock time for Ma$^{\text{GP}}$ is between 17 min and 31 min, and DHGP around 4 min (including logging, file-saving, etc.). Note that while DHGP uses optimized libraries, Ma$^{\text{GP}}$ in its current form is used as originally implemented by Ma et al. [24].

Importantly: While additional compute would certainly enable larger-scale experiments, our current setup demonstrates that even with a limited number of data points and resources, corresponding to reduced evaluation frequency and faster training, the Ma$^{\text{GP}}$ model performs effectively.

## Q    Additional Discussion

**Data Relationships.**   The data relationships in the paper mimic more closely Ma$^{\text{GP}}$, because they not only have a hierarchical structure, but also exhibit correlations among tasks. As discussed in the main paper, we expect this to be particularly well-suited for Ma$^{\text{GP}}$, a hypothesis that is strongly supported by the obtained experimental results.

**Key Advantage of Scaling Law Prediction.**   In Section 5.1, Table 3, we present results illustrating how closely the scaling laws predicted via our Monte Carlo-based approach approximate the original scaling laws obtained through conventional methods, namely, by fitting the scaling function to the full dataset. For our evaluation, we use the AbC measure. The key advantage of our approach lies in significantly reduced computational cost, while still achieving close alignment with the original scaling law. (Note that our test sets contain the most expensive learning curves.) This outperforms simple fitting to training data as it provides more support to fit the scaling law in the compute region of interest.

**Range of Models Considered for Scaling Law Experiments.**   Due to computational constraints, we focused our setup on training models for one epoch using the FineWeb-Edu dataset. Our model sizes range from 51M to 1.5B parameters, aligning with the scale used in Kaplan et al. [4], who explored models from 768M to 1.5B parameters. We therefore consider this range representative for our investigation.

**Hypothesis on Hierarchical Structures.** To support our hypothesis that hierarchical structures exist and can be exploited in learning curve datasets, we provide extensive experiments across multiple domains, including bilingual and multilingual translation tasks. To the best of our knowledge, our work is the first to investigate bi-level hierarchies for zero-shot learning curve prediction and extrapolation.

