# OpenReview forum: "Zero-Shot Performance Prediction for Probabilistic Scaling Laws"
_NeurIPS.cc/2025/Conference — NeurIPS 2025 poster_

### Official Review · Reviewer_Mugk · 2025-07-02

**Clarity:** 3
**Significance:** 3
**Originality:** 3
**Rating:** 5
**Confidence:** 3

**Summary:**

The paper proposes using latent variable multi-output Gaussian processes (specifically the MaGP model) to predict training and learning curves in a zero-shot manner. The key insight is modeling these prediction tasks as hierarchical multitask learning problems, where correlations across tasks and within hierarchies can be exploited. They demonstrate this on three NLP datasets: nanoGPT training curves, bilingual translation learning curves, and multilingual translation performance. An active learning component helps reduce uncertainty in predictions.

**Questions:**

NA

**Ethical Concerns:**

["NO or VERY MINOR ethics concerns only"]

**Final Justification:**

The authors' response has addressed my remaining concerns. I will keep the score.

**Limitations:**

Yes

**Quality:**

4

**Strengths And Weaknesses:**

Strengths

- The motivation is solid - developing scaling laws is indeed computationally expensive, and any principled method to reduce these costs would be valuable.  I think the probabilistic nature of the approach is a clear advantage over deterministic scaling law methods. Getting uncertainty estimates on scaling law parameters could be quite useful for decision-making about model development. The two-layer hierarchy matches intuitive structure in all three tasks and lets the GP share signal efficiently.
- The experimental evaluation is reasonably comprehensive given the scope. Testing on three different types of NLP tasks (language modeling, bilingual, and multilingual translation) shows some generality. The comparison against multiple baselines, including BNNs with different basis functions, helps establish the method's effectiveness. I also liked the active learning analysis showing how uncertainty can be reduced through strategic querying.
- Table 2 and Figure 7 don’t just report accuracy; they tag each train-test split or query strategy with the PetaFLOPs burned to obtain it. The active-learning policy slices AbC by roughly one-half relative to a Largest-First schedule while staying a good 2× cheaper than the Random schedule .



Weaknesses

- I'm curious about the computational costs of the MaGP method itself. While the goal is to reduce the cost of obtaining scaling laws, how expensive is the GP inference, especially with the hierarchical structure and active learning?
- The active learning results are promising, I'd also like to see more analysis of which queries are actually selected. Maybe the patterns might provide insights into what information is most valuable for scaling law prediction.
- The writing is quite dense in places, particularly sections 3 and 4. I found myself re-reading several passages to understand the model setup. Given that this is targeting a NeurIPS audience who might not all be familiar with hierarchical GPs, clearer exposition would help.

---

> ### Author Rebuttal · Authors · 2025-07-31
>
> Thank your for your supportive feedback! We address your questions individually in the following:
>
> **[W1] - Model complexity and runtime**:
>
> *Algorithmic Complexity:*
> MaGP is implemented using the inducing points method. MaGP is originally referred to as HMOGP-LV (Ma et. al., 2023). HMOGP-LV  is derived from LVMOGP (Dai et al., 2017). The computational complexity is
> $$
> \mathcal{O}\left(\max(NR, M_H) \cdot \max(D, M_X) \cdot \max(M_H, M_X)\right),
> $$
> with $N$ being the available data points per learning curve, $R$ being the number of learning curves per task, $D$ being the number of tasks, $M_X=M_r\times R$ for $M_r$ being the number of inducing input points in the r-th learning curve of a task and $M_H$ being the number of inducing output points.
>
> *Runtimes:*
> MaGP was trained and evaluated on an Intel(R) Core(TM) i7-8565U CPU.
>
> $\rightarrow$ nanoGPT-Large experiments: The wallclock time for MaGP in this scenario (w/o active learning) is between 17 min and 31 min, which includes logging and saving files. It is worth noting that MaGP is not (yet) optimized in terms of speed.
>
> $\rightarrow$ The 'active learning' component only adds negligible overhead (in comparison to the other query strategies), as the only additional operation is the computation of the variance across the already-predicted curves (which uses an optimized library).
>
> ---
> **[W2] - Patterns of Query Selection in Active Learning:**
> We thank you for raising this question. This was indeed of great interest to us as well.
> $\rightarrow$ The paper has already a corresponding figure included in Appendix F, and we will ensure that a proper reference to this is added in the main paper. (Or we might move this to the main paper to provide further insights if space permits.)
>
> ---
> **[W3] - Partially dense writing, esp. around hier. GPs:**
> Thank you for drawing our attention to this! We have already expanded the relevant sections and now provide more detailed explanations to improve clarity and completeness, as well as some underlying intuitions to cater for a wider audience.
>
> ---
> ---
>
> We would like to thank you again for the support of our work, and hope our answers have clarified your questions.
> If any further queries remain, please let us know and we will do our best to promptly address them.

---

> > ### Comment · Reviewer_Mugk · 2025-08-07
> > **Response to the authors**
> >
> > Thanks for your response. I still broadly support the idea and contribution, and I will keep the score unchanged.

---

> > > ### Author Response · Authors · 2025-08-07
> > > **Thanks for your feedback and support**
> > >
> > > Thank you again for your feedback and for supporting our work!

---

### Official Review · Reviewer_bTJp · 2025-07-03

**Clarity:** 2
**Significance:** 2
**Originality:** 2
**Rating:** 4
**Confidence:** 3

**Summary:**

The paper frames learning-curve and training-curve extrapolation in NLP as multi-task, two-level hierarchical regression problem and instantiates it with the latent-variable multi-output Gaussian process of Ma et al. (2023) (“MaGP”). Main claims are as follows :

1. Zero-shot prediction of unseen learning curves is feasible by leveraging task-level correlations and hierarchical priors .
2. The GP posterior can be Monte-Carlo–sampled to yield probabilistic scaling laws, giving uncertainty bands on fitted γ-laws .
3. By employing an active learning approach, the uncertainty of the predictions can be significantly reduced
4. The approach generalises beyond scaling-law data (nanoGPT) to bilingual and multilingual translation curves

The approach outperforms DHGP, two BNN variants, and a naïve regression baseline across MSE/MNLPD metrics. The empirical evidence relies on three small datasets (≤ 30 curves per domain) with only few points per curve, yet MaGP consistently beats baselines.

**Questions:**

- GP cubic complexity is O(N³); the paper omits runtime or inducing-point strategies. Please report wall-clock time and memory for nanoGPT-large fits.
- Would cost-aware fits (e.g., loss/FLOP) further improve results?
- Authors should try to incorporate the irreducible loss in these predictions
- For BLEU/ChrF tasks only MSE/MNLPD are shown; confidence intervals overlap heavily. Maybe break it down further and improve the plotting?

**Ethical Concerns:**

["NO or VERY MINOR ethics concerns only"]

**Final Justification:**

After authors response some of the concerns have been addressed. However I am still not strongly recommending an accept since there remains concerns if the framework is scalable and applicable to actual LLM model scaling ladder approaches.

**Limitations:**

Yes

**Quality:**

2

**Strengths And Weaknesses:**

### Strengths

- Casting curve extrapolation as hierarchical multitask GP clearly articulated.
- Folding model uncertainty into γ-law fits via MC integration is principled and valuable for budgeting decisions in LLM training.
- Active-learning cost analysis a realistic compute/accuracy trade-off, absent from most scaling-law studies .
- Comprehensive baselines, Includes DHGP, two BNN variants, and task-averaging heuristics, isolating the gain from hierarchy + latent correlation.

### Weaknesses

- All experiments use very few curves; scalability to realistic hyper-parameter spaces (≥ 10²−10³ curves) is unclear. LLM training would often depend on multiple hyperparameters and this is a combinatorially complex space
- Does not discuss vertical offset and irreducible loss, how would that impact these predictions?
- Variance-only ignores compute-cost heterogeneity across model sizes.
- Query budget is ~25; realistic large-model experiments may allow far fewer queries.
- Hierarchical grouping is hand-crafted. How sensitive is the approach to mis-specified hierarchies?

---

> ### Author Rebuttal · Authors · 2025-07-31
>
> We thank you for your helpful suggestions, and address your points in the following.
>
> **[W1] - Scaling to many hyperparameters:**
> At present, our model considers only the number of embedding parameters and the number of layers as structural inputs/hyperparameters.
> Due to computational constraints, we were limited to using 11 datapoints across 36 learning curves, which was the maximum we could accommodate on our available CPU resources (RAM-limited).  (see [Q1] below for model complexity)
>
> $\Rightarrow$ Importantly: While additional compute would certainly enable larger-scale experiments, our current setup demonstrates that even with a limited number of datapoints and resources -- corresponding to reduced evaluation frequency and faster training -- the MaGP model performs effectively!
>
> ---
> **[W2] - Vertical offset and irreducible loss:**
> Vertical offset is not explicitly modeled, as all learning curves considered have the same number of layers vertically and an identical number of datapoints. Additionally, we do not incorporate the irreducible loss term in our scaling law analysis, following the frameworks established by Kaplan *et al.* and Hoffmann *et al.*. While the irreducible loss is addressed in Hennighan’s work, incorporating it represents a natural and compelling extension for future research.
>
> $\rightarrow$ Please feel free to reach out if you require further clarification or additional details.
>
> ---
> **[W3] - Variance-based Active Learning:**
> Thank you for raising this point. You are correct that the variance-based active learning strategy explored in our work focuses on reducing the uncertainty, independent of associated compute costs. In the revised version of the paper, we therefore additionally provide AbC over cumulative compute cost to reflect the strategies' performances when computational resources are constrained.
>
> In brief: E.g. 17 queries for 'Smallest First' are roughly equivalent to only 3 for 'Largest', 7 'Active' \& 10 Random;
> $\rightarrow$ This changes the ranking based on AbC metric to prefer the 'Smallest First', followed by 'Active' and 'Random', while 'Largest' stays least competitive.
>
> Note that there exist many different ways how the cost-function in the active learning process could be structured, but this is outside the scope of this paper. We see the design of cost-terms that consider trade-offs like compute-vs-uncertainty-reduction as an interesting direction of future work.
>
> ---
> **[W4] - Query budget:**
> We thank you for raising this concern. In total, we have 25 learning curves available for querying -- however, this is the 'upper limit' that is considered in our experiments.
> Note that Figure 7 illustrates the performance when using fewer queries, demonstrating that after querying for example just 14 curves, the AbC value already falls below 0.25.
>
> ---
> **[W5] - Hierarchical Grouping:**
> We are slightly unsure what you mean with 'mis-specified' in the context of model hyperparameters.
> However, we have further validated our hypothesis that hierarchies are indeed exchangeable (and therefore indicating robustness against potential mis-specification).
> $\Rightarrow$ For the new experimental results with exchanged (swapped) hierarchy, please see the tables provided in the response to reviewer *QuFj - [W1]*.
>
> ---
> **[Q1] - Wallclock times and memory:**
> - MaGP and DHGP were trained and evaluated on an Intel(R) Core(TM) i7-8565U CPU.
> - MaGP follows the same inducing point strategy as suggested in the original work in Ma et al., by inducing every second point. (Using every point increased the runtime but did not improve on the negative log likelihood optimization criterion.)
> - The wallclock time for MaGP is between 17 min and 31 min, and DHGP around 4 min (including logging, file-saving, etc.); Note that DHGP uses optimized libraries, while MaGP in its current form is not yet optimized.
>
> ---
> **[Q2] - Cost-aware fits:**
> We are slightly unsure if we correctly understand your question. If your question refers to an active learning approach that jointly considers e.g. variance-reduction and compute-cost, this is discussed in [W3] -- and we consider this a natural and promising extension of our work for future consideration.
>
> $\rightarrow$ Please feel free to reach out if you require further clarification or additional details, or if your question has not yet been fully addressed.
>
> ---
> **[Q3] - Irreducible loss:**
> As the analyses in our paper are based on Kaplan et al.'s and Hoffmann et al.'s works, we have not incorporated the irreducible loss term in our scaling law analysis. While the irreducible loss is addressed in Hennighan’s work, we see this as a natural and compelling extension for future research -- but outside the scope of this current work.
>
> ---
> **[Q4] - Improved plot quality - BLUE/ChrF:**
> Thank you for pointing this out. We included appropriate legends and present separate performance figures for each metric in the revised version of the paper.
>
> ---
> ---
> We hope our answers have clarified all your questions.
> If you have any further queries, please let us know and we are happy to answer them.

---

> ### Author Response · Authors · 2025-08-08
> **Discussion phase nearing its end**
>
> Dear Reviewer *bTJp*,
>
> As the discussion period ends in one day, we’d be glad to hear whether our responses in the rebuttal have addressed your concerns.
>
> If our clarifications have helped resolve your concerns, we’d of course appreciate your reconsideration of the rating, if you feel that’s appropriate.
>
> In case you have any remaining questions or comments, please don’t hesitate to let us know — we’d be happy to clarify them while time allows.

---

### Official Review · Reviewer_FUMJ · 2025-07-03

**Clarity:** 3
**Significance:** 3
**Originality:** 2
**Rating:** 4
**Confidence:** 2

**Summary:**

This paper presents a framework for predicting learning and training curves of NLP models that leverages latent variable multi-output Gaussian process models. The key idea behind the approach is to model correlations between tasks and hierarchies to facilitate predictions for new configurations of models (ie with different numbers of parameters or different languages). Experiments validate the framework in three settings: scaling law prediction and predicting zero-shot performance/few-shot performance of translation models.

**Questions:**

- Line 232: Why do authors subsample to 11 datapoints? How much more computational demand does it require to use the full dataset?
- Why is there no direct comparison to prior scaling law estimation methods (e.g., those used by Kaplan et al., Hoffmann et al., Henighan et al.)?

**Ethical Concerns:**

["NO or VERY MINOR ethics concerns only"]

**Final Justification:**

My main concerns have been addressed with the authors' promise to more thoroughly introduce datasets, along with their explanations of how the experiments in this work compare to prior works. In particular, they have justified their choices of model sizes/datasets and clarified that they compare to prior methods.

**Limitations:**

yes

**Quality:**

2

**Strengths And Weaknesses:**

Strengths:
- The idea of using active learning for model prediction is creative, and results show that this method performs comparably to using the largest first strategy while reducing computational costs.
- Experiments show the broad applicability of the proposed method for developing scaling laws, selecting model architectures, and predicting performance for bilingual translation models.

Weaknesses:
- There are no comparisons to more recent scaling law estimation methods, despite them being cited in the introduction.
- The experiments are with limited models and datasets, and it is therefore unclear how robust the results are. For example, for the scaling laws experiments, the paper subsamples from the NanoGPT dataset and only evaluates language models in this previous data, which were not trained for many steps (and not larger model families).
- The datasets used in the paper are never properly introduced; authors should describe upfront what the datasets they use are and where they come from.

Minor Notes:
- Line 263: Figure has "??"
- Line 372: "proof that" --> "prove that"

---

> ### Author Rebuttal · Authors · 2025-07-31
>
> We thank you for your feedback, and address your points in the following.
>
> **[W1] - Comparison to recent scaling law estimation methods:**
> Recent scaling law experiments typically predict the scaling law displayed in 'blue' in Figure 2 by utilizing the full available dataset (i.e. expensive training of all models), without employing zero-shot prediction to reduce computational complexity/cost.
> $\rightarrow$ In this sense, these experiments are effectively represented by our baseline 'ground-truth' scaling law, to which we compare our results.
>
> ---
> **[W2] - Limited Models and Datasets:**
> Thank you for raising this concern. Due to computational constraints, we focused our setup on training models for one epoch using the FineWeb-Edu dataset. Our model sizes range from 51M to 1.5B parameters, aligning with the scale used in Kaplan et al., who explored models from 768M to 1.5B parameters. We therefore consider this range representative for our investigation.
>
> To support our hypothesis that hierarchical structures exist and can be exploited in learning curve datasets, we provide extensive *experiments across multiple domains*, including *bilingual* and *multilingual translation tasks*. To the best of our knowledge, our work is the first to investigate bi-level hierarchies for zero-shot learning curve prediction and extrapolation.
>
> ---
> **[W3] - Introduction of Datasets:**
> While the datasets we are using are introduced in Section 4, we agree that the detailed composition and origin is somewhat unclear;
> $\rightarrow$ We have added an extended Section describing their creation as well as inherent properties to the appendix of the paper (including references in the main text). Note that all these datasets were created by us.
>
> ---
> **Re: Minor Notes** -- We thank you for bringing these to our attention; textual and grammatical errors have been corrected.
>
> ---
> **[Q1] - Subsampling & computational demand:**
> > Line 232: Why do authors subsample to 11 datapoints? How much more computational demand does it require to use the full dataset?
>
> Due to computational constraints, we were limited to using 11 datapoints across 36 learning curves, which was the maximum we could accommodate on our available *CPU* resources (RAM-limited).
>
> Regarding model complexity:
> - MaGP is implemented using the inducing points method.
> - MaGP is originally referred to as HMOGP-LV (Ma et. al., 2023). HMOGP-LV  is derived from LVMOGP (Dai et al., 2017).
>
> The computational complexity is
> $$
> \mathcal{O}\left(\max(N R, M_H) \cdot \max(D, M_X) \cdot \max(M_H, M_X)\right),
> $$
> with $N$ being the available data points per learning curve, $R$ being the number of learning curves per task ($d$), $D$ being the number of tasks $t$, $M_X=M_r\times R$ for $M_r$ being the number of inducing input points in the r-th learning curve of a task and $M_H$ being the number of inducing output points.
>
> $\Rightarrow$ Importantly: While additional compute would certainly enable larger-scale experiments, our current setup demonstrates that *even with a limited number of datapoints and resources* -- corresponding to reduced evaluation frequency and faster training -- the MaGP model *performs effectively*!
>
> ---
> **[Q2] - Comparison to prior work:**
> > *Why is there no direct comparison to prior scaling law estimation methods (e.g., those used by Kaplan et al., Hoffmann et al., Henighan et al.)?*
>
> As stated above in [W1], recent scaling law experiments typically predict the scaling law displayed in 'blue' in Figure 2 by utilizing the full available dataset (i.e. expensive training of all models), without employing zero-shot prediction to reduce computational complexity/cost.
> $\rightarrow$ In this sense, these experiments are effectively represented by our baseline 'ground-truth' scaling law, to which we compare our results.
>
> ---
> ---
> We hope our answers addressed all your questions.
> If you have any further queries, please do not hesitate to reach out.

---

> ### Author Response · Authors · 2025-08-08
> **Discussion phase approaching its end**
>
> Dear Reviewer *FUMJ*,
>
> As the discussion period ends in *one day*, we’d be glad to hear whether our responses in the rebuttal have addressed your concerns.
>
> If our clarifications have helped resolve your concerns, we’d of course appreciate your reconsideration of the rating, if you feel that’s appropriate.
>
> In case you have any remaining questions or comments, please don’t hesitate to let us know — we’d be happy to clarify them while time allows.

---

> ### Comment · Reviewer_FUMJ · 2025-08-08
>
> I thank the authors for their thorough response. I believe my main concerns have been addressed with the authors' promise to more thoroughly introduce datasets, along with their explanations of how the experiments in this work compare to prior works. I would encourage authors to explicitly clarify [W1] in the paper. I have raised my score accordingly.

---

> > ### Author Response · Authors · 2025-08-09
> > **Thanks for your feedback and support**
> >
> > Thank you again for your feedback, we've already taken your suggestions into account while revising our manuscript.
> >
> > Thanks for supporting our work!

---

### Official Review · Reviewer_QuFj · 2025-07-09

**Clarity:** 3
**Significance:** 2
**Originality:** 3
**Rating:** 4
**Confidence:** 3

**Summary:**

The paper applies hierarchical Gaussian processes, often used in other types of hierarchical datasets, to the task of predicting learning curves (LCs) for NLP models and estimating frontier scaling laws for these LCs.

Hierarchical Gaussian processes encode outside knowledge/belief about relationships/correlations between tasks which permit stronger few-shot modeling or zero-shot modeling in LCs from tasks without much data.

The study is predominantly empirical, examining LC prediction on different small models while tweaking two model or dataset parameters (e.g. size of embedding AND number of hidden layers, or target language AND source language of translation). Hierarchical GPs permit structuring the modification of one parameter as a subtask of the supertask of modification of the other parameter.

Using a GP from Ma et al. [1], the authors explore LC prediction across a number of tasks and zero-shot few-shot settings. They also explore the error in scaling laws (linear frontier functions in log-log space) estimated from LCs. The authors compare to baselines including other GPs, Bayesian Neural Networks, and naive baselines such as averaging across tasks that are known to exist in the same subgroup in the hierarchy.

Results indicate that good performance prediction and scaling law estimation can be made with Hierarchical GPs, and furthermore, that using uncertainty in probabilistic prediction can be used to select candidates for querying LCs in active learning.

**Questions:**

Most of my questions are actually covered in my W1 and W2. I am very curious about a deeper analysis of many of the experiments and arguments made.

I have only one more question to write here that has not yet been covered.

**Q1**

In Figure 7, can you include a curve with cumulative compute cost as the x axis? I wonder if Smallest First is still competitive if you consider the fact that Smallest First is much cheaper to get to a critical mass of queries as compared to any other method.

**Ethical Concerns:**

["NO or VERY MINOR ethics concerns only"]

**Final Justification:**

Note: August 6, 2025 update: Score increased from 2->4 after author rebuttal. Final decision pending discussion with other reviewers. Quality 2->3, Clarity 1->3.

**Limitations:**

Limitations section is a bit short. It mostly discusses limitations on the scope and breadth of analysis. I am curious about the limitations of this methodology. For example, which relationships will Gaussian processes struggle to handle? What are the failure modes of MaGP that DHGP or other models don't suffer from?

As far as societal impacts go, I do not think there are major adversary societal impacts to be concerned with. Many techniques have been proposed or already are in use for scaling law estimation. Generally, being aware of scaling laws helps us plan to improve efficiency and reduce waste while training models.

**Quality:**

3

**Strengths And Weaknesses:**

Strengths

**S1**

Novel application of this model to my knowledge.

**S2**

Wide applicability, and selection of hierarchical modeling for LC prediction is an insightful way to apply belief about relationships in LCs.

Weaknesses

**W1**

The paper suffers a lack of clarity in communicating its assumptions and how they map to modeling decisions. A one-off statement in Section 5 that "If correlations between data samples of different tasks are not relevant, or even a hindrance, this baseline is supposed to outperform the MaGP model" but provides no further explanation of this distinction or the theoretical basis for the claim. I think extra attention should be paid to explaining a) the hypothesis about the specific types of hierarchical relationships that DHGP and MaGP can each express and b) why all of the data relationships in the paper more closely mimic MaGP and not DHGP.

As another example, another one-off statement in 5.3 mentions that "hierarchies are indeed exchangeable", which I interpret to mean that modeling one parameter axis as the higher level and the other as the lower, or vice versa, leads to no difference in effect. Yet what results indicate this? How does this interact with our chief assumptions? Are hierarchies exchangeable for all examined problems?

In essence I am requesting additional focus on the core hypothesis, a categorization of what evidence would contribute toward or against it, a clear set of research questions that are being examined, and clearly formatted evidence to address each research question.

**W2**

The scope of experiments is relatively wide, but the depth in analysis, explanation, and experimentation granted each experimental setting leaves some to be desired. For example, why pursue a different set of baselines in each approach? Is there a justifiable reason, is it convenience, etc.? Each experimental setting is introduced technically but not in enough detail to describe these sorts of decisions.

The haphazard use of baselines can harm my assessment of the significance. I have no reason to believe right now that the paper captures a sharp enough view of the SoTA in performance prediction for each task type that the integration of the MaGP method actually means something substantial for those practitioners interested in predicting LCs.

As another example of some missing depth of analysis, the process of using Monte Carlo estimation and then fitting a scaling law to the estimates in addition to the real data is interesting but worth giving a full treatment or exploration. Why does bootstrapping a GP and then fitting a linear law tend to outperform merely fitting the linear law to the existing data? By how much does it do so? Is there any justification as to why this would be better?

**W3**

Several figures suffer from clarity issues. For example:

Figure 4 Left -- how is the main diagonal modeled as test data if the test set is actually the next diagonal over to the right?

Figure 4 Left -- No legend, no axes labels, no sense of what conclusions to take away.

Figure 4 Right -- Font is too small to read, unclear what red vs blue numbers are, legends poorly formatted.

Figure 8 -- no legend.

Figure B.5 -- hard to read.

**W4**

Tables suffer from clarity issues. For example:

Convention states that a bold value is used to indicate the winner of a head-to-head competition in tables. Yet sometimes the bold values are inappropriately applied. For example:

Table E.2 -- no bolding when MaGP does not win?

Table H.3 -- bolding of non-winning values?

Use of arrows on column headers to indicate direction of optimal metric values inconsistent. Present, for example, in Table 2 but not Table 1.

**Grammatical and textual nitpicks**

82 -- no space before “Recently”

178 -- (512-8 two times?)

263 -- Figure ??

267 -- "weather" -> whether

281 -- Flops exponents formatting off

294 --"while the compute costs" -- incomplete phrase

296 -- "weather" -> whether

336 -- "hense"

364 -- sentence has no following punctuation

372 -- proof -> prove

373 -- "baselines"

Appendix A Line 33 -- grammar needs a revision

Appendix B Line 72 -- "increaseing"

Appendix D Line 122 -- "commitee" & 123 "leanring"

---

> ### Author Rebuttal · Authors · 2025-07-31
>
> We thank you for your constructive review, and address your points individually in the following:
>
> **[W1] - Assumptions and Modeling decisions:**
> We thank you for bringing this to our attention!
>
> > *I think extra attention should be paid to explaining a) the hypothesis about the specific types of hierarchical relationships that DHGP and MaGP can each express*
>
> $\rightarrow$ MaGP introduces an additional latent variable $\mathbf{h}$ to model correlations among tasks $t$, allowing it to capture cross-task dependencies.
>
> $\rightarrow$ In contrast, DHGP does not rely on this shared latent representation. Instead, it leverages a hierarchical modeling approach defined as:
> $$
> \begin{aligned}
> f(x) &\sim \mathcal{GP}(0, k_f(x, x')) \\\\
> g(x) &\sim \mathcal{GP}(f(x), k_g(x, x')) \\\\
> l(x) &\sim \mathcal{GP}(g(x), k_l(x, x')) \\\\
> y(x) &= l(x) + \varepsilon, \quad \varepsilon \sim \mathcal{N}(0, \sigma^2)
> \end{aligned}
> $$
> Following and inspired by the approach of Hensman et al. [27], we interpret $f$ as a general function capturing global learning curve trends across tasks. The next level, $g$, captures variations due to a coarser factor, i.e., the number of embedding dimensions. At the lowest level, $l$ models individual learning curves, which vary with the number of layers.
>
> $\Rightarrow$ DHGP is therefore particularly suited for discovering clusters in the data, where each cluster can be thought of as a group of tasks sharing similar characteristics (e.g., number of embedding parameters).
> $\Rightarrow$ This is in contrast to MaGP, which assumes that inter-task correlations exist and can be captured through a latent variable $\mathbf{h}$.
> $\Rightarrow$ If such correlations are weak or non-existent, we expect the hierarchical structure of DHGP to provide a more appropriate inductive bias than MaGP, allowing it to learn within-cluster trends effectively.
>
> > *[...] b) why all of the data relationships in the paper more closely mimic MaGP and not DHGP.*
>
> The data relationships in the paper mimic more closely MaGP, because they not only have a hierarchical structure, but also exhibit correlations among tasks. As discussed above, we expect this to be particularly well-suited for MaGP -- a hypothesis that is strongly supported by the obtained experimental results.
>
> ---
> > *[...] "hierarchies are indeed exchangeable", which I interpret to mean that modeling one parameter axis as the higher level and the other as the lower, or vice versa, leads to no difference in effect. [...] Are hierarchies exchangeable for all examined problems?*
>
> To provide experimental evidence towards the hypothesis of exchangeable hierarchies (everything else kept identical), we provide the following additional results:
>
> 1. Scaling law experiments (layer-over-embedding):
> | Method | TT | MSE | MAE | MNLPD |
> |-|-|-|-|-|
> | MaGP | Quad | 0.04 ± 0.04 | 0.11 ± 0.07 | 1.82 ± 2.23 |
> | DHGP  | Quad | 0.11 ± 0.003 | 0.26 ± 0.01| 0.34 ± 0.02 |
> |-|-|-|-|-|
> | MaGP | Tri | 0.02 ± 0.01 | 0.09 ± 0.03 | 0.37 ± 0.79 |
> | DHGP | Tri | 0.13 ± 0.05 | 0.29 ± 0.06 | 0.48 ± 0.16 |
> |-|-|-|-|-|
> | MaGP | T1 | 0.01 ± 0.02 | 0.08 ± 0.04 | -0.88 ± 0.58 |
> | DHGP | T1 | 0.16 ± 0.02 | 0.34 ± 0.02 | 1.18 ± 0.43 |
>
> 2. Bilingual experiments (target-to-source)
>  | Method | TT | MSE  | MAE  | MNLPD |
> |-|-|-|-|-|
> | MaGP | mBart chrF | 43.24 ± 13.72 | 5.297 ± 0.87 | 3.698 ± 0.43 |
> | DHGP | mBart chrF | 127.53 ± 0.10 | 8.3 ± 0.00 | 7.44 ± 0.07 |
> | MaGP | mBart BLEU | 13.287 ± 4.41 | 2.625 ± 0.45 | 5.15 ± 2.43 |
> | DHGP | mBart BLEU | 33.17 ± 0.22 | 3.835 ± 0.01 | 4.26 ± 0.14 |
> |-|-|-|-|-|
> | MaGP | Transf chrF | 18.801 ± 3.32 | 3.08 ± 0.22 | 2.866 ± 0.14 |
> | DHGP | Transf chrF | 35.45 ± 0.02 | 4.15 ± 0.00 | 4.34 ± 0.04 |
> | MaGP | Transf BLEU | 1.018 ± 0.23 | 0.53 ± 0.86 | 2.06 ± 0.48 |
> | DHGP | Transf BLEU | 4.577 ± 0.00 | 1.08 ± 0.0 | 4.62 ± 0.16 |
>
> 3. Architecture design experiments:
> Already provided in Section 5.3, Figure 9.
>
> $\Rightarrow$ All results provide strong evidence that the hierarchies across all tasks examined in our work are indeed exchangeable without sacrificing performance.
>
> ---
> > *In essence I am requesting additional focus on the core hypothesis, a categorization of what evidence would contribute toward or against it [...].*
>
> *Core Hypothesis*:
> We hypothesize that learning curve (LC) datasets exhibit hierarchical structure, specifically bi-level hierarchies.
> $\rightarrow$ We perform a range of experiments to provide insights and evidence towards this core hypothesis by analyzing the respective performance of MaGP vs DHGP.
>
> Additional background info: As previously discussed, if we assume that clusters existed in the dataset, DHGP would be the preferred model, as it is capable of discovering and modeling such latent structure. On the other hand, if the dataset's underlying structure exhibits strong inter-task correlations and a hierarchical structure within each task, then MaGP are expected to outperform.
>
> *Additional hypothesis*: The exact ordering of bilevel hierarchies present in this data is exchangeable.
> $\rightarrow$ As stated above, our newly-included experimental results provide evidence that supports this hypothesis.
>
> *Final hypothesis*: The model which best captures the structure of learning curve datasets can be successfully used for scaling law extrapolation, due its good zero-shot performance.
> $\rightarrow$ This hypothesis is empirically supported in Subsection 5.1.
>
> ---
> **[W2.1&2] - Experiments and different baselines:**
> We have reworked the experiment section to provide more explanations for our choice of baselines across all experiments. In brief:
>
> $\rightarrow$ The necessity for different baselines arises primarily from the different nature of the datasets and the way learning curves are structured in each setting.
>
> - For the nanoGPT dataset, we have access to a sufficient number of learning curves and data points, making a Bayesian Neural Network (BNN) a valid and effective baseline. This choice is further motivated by prior work, such as Klein et al., where BNNs were successfully applied for learning curve extrapolation and zero-shot prediction.
>
> - In contrast, applying a BNN in the bilingual translation setting is more challenging. While we explored modeling the learning curves as a function of source and target language pairs, the BNN struggled to generalize and capture the upward trend in learning curves. Also note, we only have $25$ learning curves for training, and each source and target language is only $5$ times represented, which given the noisy nature of this dataset (see Figure 4 left) is not enough training data to perform well.
>  $\rightarrow$ Given the assumption that source and target languages may share structural similarities, we instead opted for a simpler and more robust baseline: averaging available learning curves across source and target languages. This approach offers a language-agnostic approximation of performance trends.
>
> - The final experiment involves an extrapolation task, where we predict learning curves from a single source (or target) language into five different target (or source) languages. Unlike the bilingual translation setup, the training grid in this scenario is defined by three distinct model sizes (see Figure 4, right). The dataset for this experiment contains only 15 LCs, which is (again) insufficient for effectively training a BNN to generalize. Hence, a regression-based baseline, such as polynomial or power-law fitting, is a more suitable and reliable choice.
>
> ---
> **[W2.3] - Monte Carlo:**
> In Section 5.1 and Table 2, we present results illustrating how closely the scaling laws predicted via our Monte Carlo-based approach approximate the original scaling laws obtained through conventional methods -- namely, by fitting the $L(C)$ scaling function to the full dataset. For our evaluation, we use the AbC measure.
> $\rightarrow$ The key advantage of our approach lies in significantly reduced computational cost, while still achieving close alignment with the original scaling law. (Note that our test sets contain the most expensive learning curves.)
> $\rightarrow$ This outperforms simple fitting to training data as it provides more support to fit the scaling law in the compute region of interest.
>
> ---
> **[W3-4] - Textual quality:**
> We have rectified all formatting and textual issues -- thank you for bringing these to our attention!
>
> ---
> **[Q1]**
> Addressed in [W1&2]
>
> ---
> **[Q2] - Compute on x-axis:**
> Thank you for this great suggestion! We have added the curves -- and this indeed provides very interesting insights!
> In brief: E.g. 17 queries for 'Smallest First' are roughly equivalent to only 3 for 'Largest', 7 'Active' & 10 Random;
> $\rightarrow$ This changes the ranking based on AbC metric to prefer the 'Smallest First', followed by 'Active' and 'Random', while 'Largest' stays least competitive.
>
> ---
> **Limitations:**
> We acknowledge a current limitation of MaGP regarding interpolation across the architectural grid. Specifically, our dataset is structured on a fixed grid defined by the number of layers and embedding parameters. As a result, the model in its current form cannot interpolate to unseen values of embedding parameters. Interpolation across unseen numbers of layers is technically feasible, but it would yield an averaged estimate between the nearest known configurations, which may not reflect true model behavior. We see this as a promising direction for future work and plan to investigate more flexible representations to address this limitation (e.g. w/ meta parameters).
>
> ---
> ---
> We thank you again for your detailed feedback, and acknowledge that framing and textual quality *did* require some polishing. We'd like to highlight that a large part of your concerns were related to these issues -- all of which have been taken care of in the revised version of our manuscript thanks to your input!
>
> We hope we have addressed your questions; if any remain, please let us know.

---

> ### Comment · Reviewer_QuFj · 2025-08-06
>
> I'm thoroughly grateful to the authors for receiving my content and presentation suggestions.
>
> All of the answers make sense to me and I find my issues addressed.
>
> I applaud the authors for their rebuttal and I now raise my score to a 4. I look forward to discussing with other reviewers any outstanding issues.

---

> ### Author Response · Authors · 2025-08-06
> **Thank you for your feedback and support**
>
> Thank you again for your constructive feedback and for supporting our work!

---

### Decision · Program_Chairs · 2025-09-17

**Decision:**

Accept (poster)

**Comment:**

The paper was reviewed by four experts who provided detailed reviews including strengths and weaknesses of the paper. The authors engaged with the reviewers in providing clear responses to the reviewers' questions. All four reviewers were satisfied with the response and raised their scores (or maintained accept recommendations). In the end the reviewers are unanimous in recommending accepting the paper and the Area Chair sees no reason to override their recommendation.